# *C. elegans* CLASP/CLS-2 negatively regulates membrane ingression throughout the oocyte cortex and is required for polar body extrusion

**Aleesa J. Schlientz**, **Bruce Bowerman** *

Institute of Molecular Biology, University of Oregon, Eugene, OR, United States of America

* bowerman@uoregon.edu

## Abstract

The requirements for oocyte meiotic cytokinesis during polar body extrusion are not well understood. In particular, the relationship between the oocyte meiotic spindle and polar body contractile ring dynamics remains largely unknown. We have used live cell imaging and spindle assembly defective mutants lacking the function of CLASP/CLS-2, kinesin-12/KLP-18, or katanin/MEI-1 to investigate the relationship between meiotic spindle structure and polar body extrusion in *C. elegans* oocytes. We show that spindle bipolarity and chromosome segregation are not required for polar body contractile ring formation and chromosome extrusion in *klp-18* mutants. In contrast, oocytes with similarly severe spindle assembly defects due to loss of CLS-2 or MEI-1 have penetrant and distinct polar body extrusion defects: CLS-2 is required early for contractile ring assembly or stability, while MEI-1 is required later for contractile ring constriction. We also show that CLS-2 both negatively regulates membrane ingression throughout the oocyte cortex during meiosis I, and influences the dynamics of the central spindle-associated proteins Aurora B/AIR-2 and MgcRacGAP/CYK-4. We suggest that proper regulation by CLS-2 of both oocyte cortical stiffness and central spindle protein dynamics may influence contractile ring assembly during polar body extrusion in *C. elegans* oocytes.

## Author summary

The precursor cells that produce gametes—sperm and eggs in animals—have two copies of each chromosome, one from each parent. These precursors undergo specialized cell divisions that leave each gamete with only one copy of each chromosome; defects that produce incorrect chromosome number cause severe developmental abnormalities. In oocytes, these cell divisions are highly asymmetric, with extra chromosomes discarded into small membrane bound polar bodies, leaving one chromosome set within the much larger oocyte. How oocytes assemble the contractile apparatus that pinches off polar bodies remains poorly understood. To better understand this process, we have used the nematode *Caenorhabditis elegans* to investigate the relationship between the bipolar structure

**Data Availability Statement:** All relevant data are within the manuscript and its Supporting Information files.

**Funding:** This work was supported by funding from the National Institutes of Health (www.nih.gov) T32GM007413 (AJS) and R01GM049869 and R35GM131749 (AJS and BB). The funders had no role in study design, data collection and analysis, decision to publish, or preparation of the manuscript.

**Competing interests:** The authors have declared that no competing interests exist.

that separates oocyte chromosomes, called the spindle, and assembly of the contractile apparatus that pinches off polar bodies. We used a comparative approach, examining this relationship in three spindle assembly defective mutants. Bipolar spindle assembly and chromosome separation were not required for polar body extrusion, as it occurred normally in mutants lacking a protein called KLP-18. However, mutants lacking the protein CLS-2 failed to assemble the contractile apparatus, while mutants lacking the protein MEI-1 assembled a contractile apparatus that failed to fully constrict. We also found that CLS-2 down-regulates membrane ingression throughout the oocyte surface, and we discuss the relationship between oocyte membrane stiffness and polar body extrusion.

## Introduction

Oocyte meiosis comprises a single round of genome replication followed by two highly asymmetric cell divisions that produce one haploid gamete and two small polar bodies that contain discarded chromosomes [1–4]. An acentrosomal spindle segregates homologous chromosomes during the first reductional division, called meiosis I, when half of the recombined homologs are extruded into the first polar body. The equational meiosis II division then segregates sister chromatids, with half extruded into a second polar body and half remaining in the oocyte cytoplasm. Despite being essential for reducing oocyte ploidy, little is known about the cues that organize and influence the actomyosin contractile ring that mediates polar body extrusion.

The dynamics of membrane ingression relative to the oocyte meiotic spindle during polar body extrusion vary from species to species, and the relationships between spindle structure and furrow ingression remain poorly understood [5]. In *Caenorhabditis elegans*, the oocyte contractile ring initially forms distal to the membrane-proximal meiotic spindle pole, with the spindle axis oriented orthogonally to the overlying cell cortex [5–7]. When observed *in utero*, the contractile ring ingresses past both the membrane-proximal pole and one set of the segregating chromosomes to then constrict and ultimately separate the nascent polar body from the oocyte [6]. These dynamics contrast substantially with mitotic cytokinesis (Fig 1A), during which signals from astral microtubules and the central spindle position the contractile ring midway between the two spindle poles [8–10]. While the signals required for contractile ring assembly and constriction during mitotic cytokinesis are relatively well understood, how oocyte meiotic spindles influence contractile ring dynamics during polar body extrusion is not known.

Several genes have been shown to be required for polar body extrusion in *C. elegans*, but how their functions are coordinated remains poorly understood. Similar to mitotic cytokinesis, polar body cytokinesis requires filamentous actin and the non-muscle myosin II heavy-chain NMY-2 and light-chains MLC-4 and MLC-5 [6,11,12]. The cytoskeletal scaffolding protein anillin/ANI-1 facilitates transformation of the initial actomyosin contractile ring into a midbody tube, with anillin depletion resulting in large and unstable polar bodies that often fuse with the oocyte [7]. Consistent with its role as a key activator of cortical actomyosin, the small GTPase RhoA (RHO-1) and its RhoGEF ECT-2 also are required for oocyte polar body extrusion [6,13]. Knockdown of the centralspindlin complex, comprised of MgcRacGAP/CYK-4 and kinesin-6/ZEN-4, results in the assembly of abnormally large contractile rings and a subsequent failure in extrusion [6]. Finally, the chromosomal passenger complex (CPC) member Aurora B/AIR-2 also is required [14].

Formation of the contractile ring distal to both meiotic spindle poles raises the question of how the ring moves relative to the spindle such that it constricts midway between segregating chromosomes. One mechanism proposed for *C. elegans* is that global contraction of

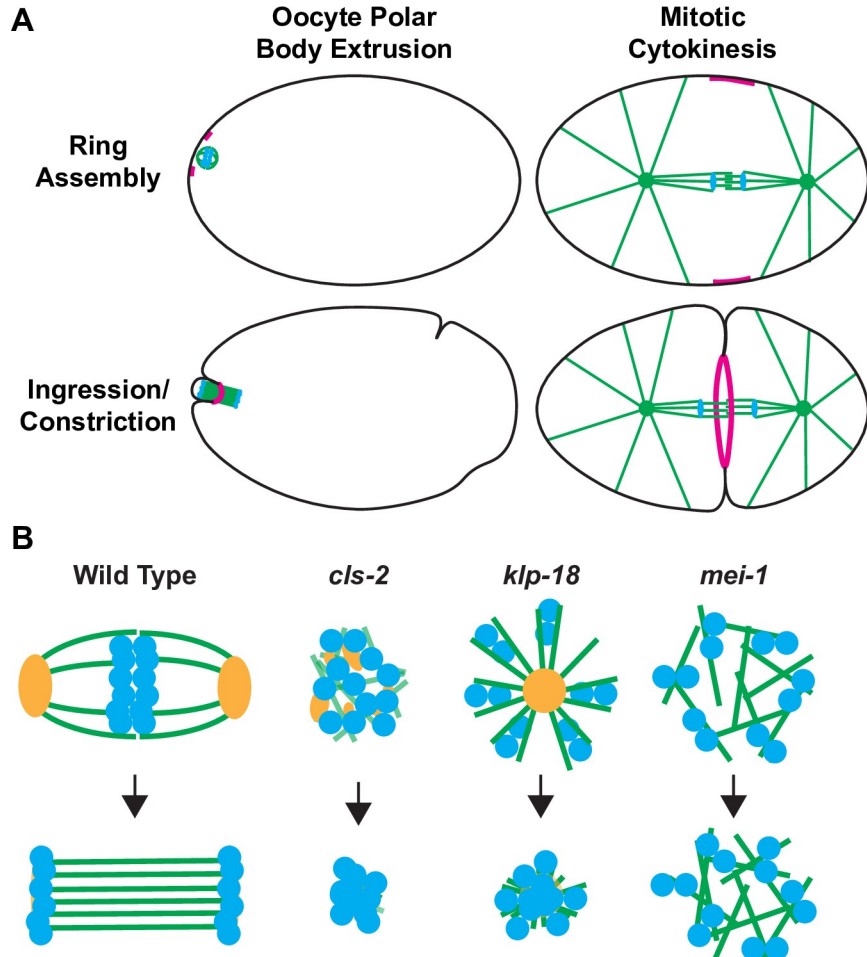

**Fig 1. Schematics of oocyte meiotic polar body extrusion, mitotic cytokinesis and oocyte meiotic spindle assembly-defective mutants.** (A) The positioning and dynamics of contractile ring assembly and ingression during oocyte polar body extrusion and mitotic cytokinesis. (B) Illustrations of oocyte meiotic spindle structure in control and mutant oocytes. Green = microtubules, blue = chromosomes, magenta = contractile rings, and orange = ASPM-1 pole marker, black = plasma membrane. See text for details.

actomyosin throughout the oocyte cortex produces a hydrostatic cytoplasmic force that, combined with depletion of cortical actomyosin overlying the membrane-proximal pole, leads to an out-pocketing of the membrane within the contractile ring that pulls the tethered spindle into the protruding pocket [6,15]. In support of this hypothesis, increased global cortical contractility due to depletion of casein kinase 1 gamma (CSNK-1), a negative regulator of RhoA activity, often results in extrusion of the entire meiotic spindle [15]. However, assessing whether global cortical contractility is required for polar body extrusion has been challenging due to the overlap in requirements for global cortical contractility and polar body contractile ring assembly and constriction.

Despite a stereotyped spatial relationship between the oocyte meiotic spindle and contractile ring assembly and ingression, little is known about how the spindle might influence ring assembly and dynamics. Nevertheless, four observations suggest that in *C. elegans*, the meiotic spindle provides important cues. First, meiotic spindles that fail to translocate to the oocyte cortex induce the formation of membrane furrows that ingress deeply towards the displaced spindle [6]. Second, loss of the centralspindlin complex, present at the central spindle during

anaphase, results in the formation of rings with an abnormally large diameter [6]. Third katanin/*mei-1* mutants, which assemble apolar spindles, produce very large polar bodies during meiosis II when microtubule severing is compromised [16]. Finally, while work in mice suggests that chromosomes themselves may provide cues for ring assembly and polar body extrusion via the small GTPase Ran [17,18], knock down of *C. elegans* RAN-1 does not prevent polar body extrusion [19,20]. These findings suggest that in *C. elegans* the oocyte meiotic spindle provides cues that influence contractile ring assembly and ingression.

To explore the relationship between meiotic spindle assembly and polar body extrusion, we have examined polar body extrusion in three spindle assembly defective mutants that each lack the function of a conserved protein: CLASP/CLS-2, kinesin-12/KLP-18, or katanin/MEI-1 (Fig 1B). CLASP family proteins promote microtubule stability through their association with microtubules and their tubulin heterodimer-binding TOG (Tumor Over-expressed Gene) domains, decreasing the frequency of microtubule catastrophe and promoting rescue of depolymerizing microtubules [21–24]. Moreover, human CLASPs have been shown to influence not only microtubule stability and dynamics, but also to interact with actin filaments and potentially crosslink filamentous actin and microtubules [25]. CLS-2 is one of three *C. elegans* CLASPs and is required for mitotic central spindle stability and for oocyte meiotic spindle assembly and chromosome segregation [26–29]. Vertebrate kinesin-12/KLP-18 family members promote mitotic spindle bipolarity by contributing to forces that push apart anti-parallel microtubules [30–32]. Consistent with such a function, *C. elegans* oocytes lacking the kinesin-12 family member KLP-18 form monopolar meiotic spindles that draw chromosomes towards a single spindle pole and fail to segregate them [33–35]. The widely conserved microtubule severing complex katanin is encoded by two *C. elegans* genes, *mei-1* and *mei-2* [36,37]. Loss of either subunit results in the formation of apolar meiotic spindles that fail to congress or segregate chromosomes [33,38,39], and mutant alleles with reduced microtubule severing extrude abnormally large polar bodies during meiosis II [16].

Here we report our use of fluorescent protein fusions and live cell imaging to characterize polar body extrusion during meiosis I in mutants lacking the function of CLS-2, KLP-18 or MEI-1. Previous studies indicate that both CLS-2 and MEI-1 are involved in polar body extrusion, with oocytes lacking CLS-2 frequently failing to extrude polar bodies [26,29,40], and oocytes lacking MEI-1 forming very large polar bodies [16,41–43]. Furthermore, *klp-18* mutants sometimes lack an oocyte pronucleus, suggesting that all oocyte chromosomes can be extruded [33]. However, the process of polar body extrusion has not been directly examined in any of these mutants. Our live imaging of contractile ring assembly and ingression in these three spindle-defective mutant backgrounds shows that bipolar spindle assembly and chromosome segregation are not required for oocyte contractile ring assembly and polar body extrusion. However, CLS-2 is required for contractile ring assembly or stability, and acts as a negative regulator of global cortical membrane ingressions, while MEI-1 may be required late in polar body extrusion for contractile ring constriction. We suggest that CLS-2 influences oocyte cortical stiffness to promote polar body extrusion, and we further suggest that disruption of central spindle-associated protein dynamics may also contribute to proper oocyte meiotic contractile ring assembly and ingression.

## Results

### CLS-2 is required for oocyte meiotic spindle assembly and polar body extrusion

To investigate the role of CLS-2, we first examined the localization of an extragenic CLS-2:: GFP fusion [27] in live oocytes (Fig 2A, S1 Fig, S1 and S2 Movies). Consistent with previous

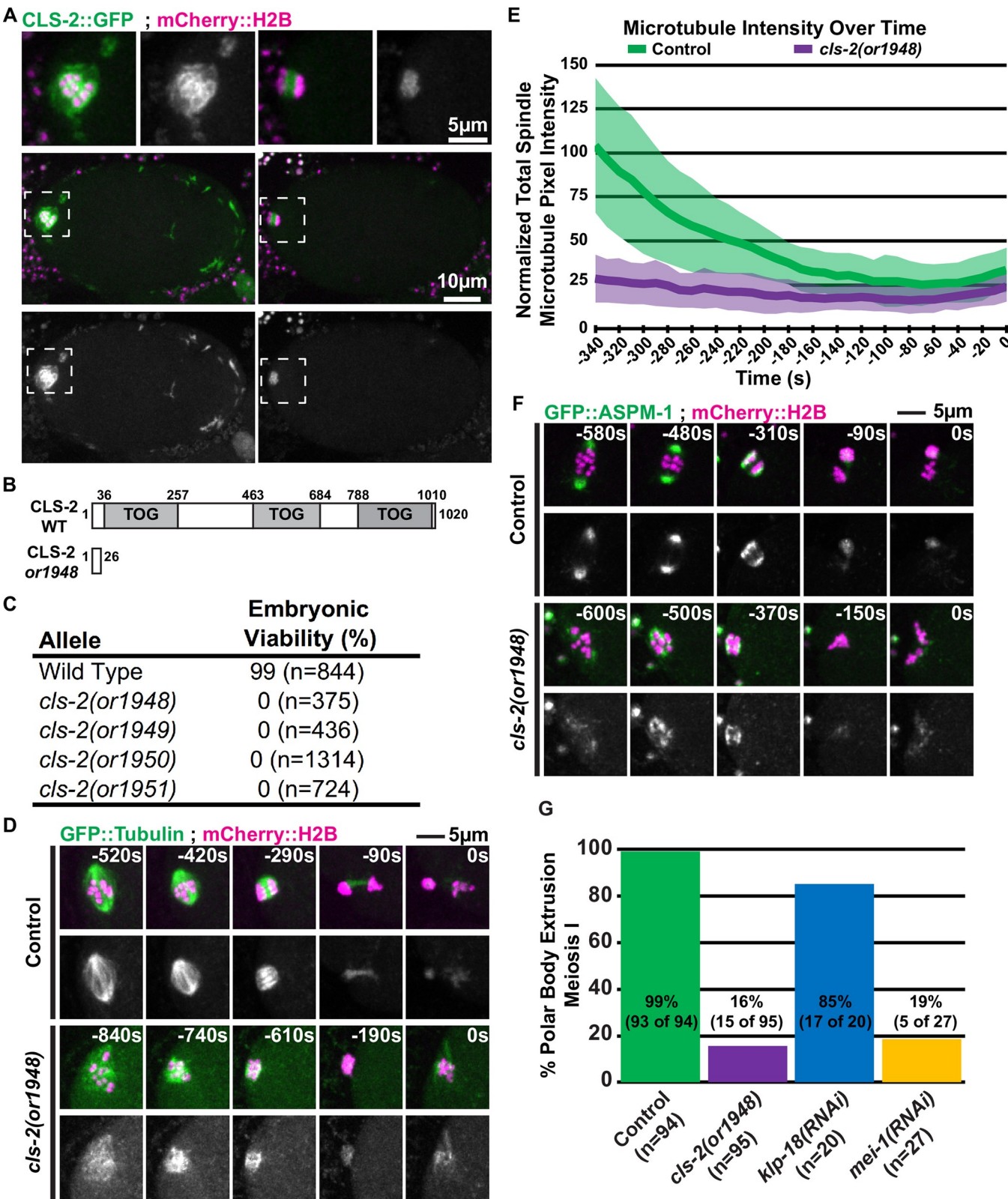

**Fig 2. CLS-2 is required for oocyte meiotic spindle assembly and polar body extrusion.** (A) Control oocytes expressing CLS-2::GFP and mCherry:H2B during meiosis I; dashed boxes indicate the zoomed-in regions shown in top row. (B) Protein domain map for wild type CLS-2 [27], and location of first premature stop codon due to frameshift in *cls-2(or1948)*. (C) Table of embryonic viability in wild type (N2) and *cls-2* CRISPR-generated loss of function alleles.

(D) *Control and mutant oocytes expressing GFP::Tubulin and mCherry::H2B during meiosis I; t = 0 seconds (0s) corresponds to the end of meiosis I and beginning of meiosis II (see* Materials and methods). (E) *Comparison of integrated spindle microtubule pixel intensity over time between control (n = 17) and cls-2 mutant oocytes (n = 9 @ -340s, n = 10 @ -330 to -300s, n = 11 @ -290 to -160s, and n = 12 @ -150 to 0s), with the average intensity indicated and one standard deviation indicated. Here and in subsequent figure panels, T = 0s corresponds to the end of meiosis I and beginning of meiosis II, unless indicated otherwise (see* Materials and methods). *The -340s time point corresponds to roughly the beginning of anaphase in control oocytes.* (F) *Control and mutant oocytes expressing GFP::ASPM-1 and mCherry::H2B during meiosis I.* (G) *Percent of control and mutant oocytes that extrude a polar body during meiosis I, as determined by mCherry or GFP-fused histone signal remaining stably ejected from the oocyte cytoplasm for the duration of imaging (see* Materials and methods).

reports, CLS-2::GFP initially localized to meiosis I spindle microtubules and kinetochore cups, and to small patches, previously described as rod-shaped structures or linear elements, dispersed throughout the oocyte cortex [26,44]. Around the time of anaphase onset, the cortical CLS-2::GFP patches disappeared and CLS-2::GFP localized to the central spindle between the segregating chromosomes [29,40]. These results suggest that CLS-2 might have roles not only at the oocyte meiotic spindle but also throughout the oocyte cortex.

Previous studies of CLS-2 requirements, using RNA interference (RNAi) or auxin-induced degradation (AID) of degron tagged CLS-2, have emphasized its central spindle function [26,29,40]. To more definitively assess its roles during oocyte meiotic cell division, we used CRISPR/Cas9 to generate putative null alleles. Each of the four alleles we isolated contains small insertions or deletions that result in frame shifts and premature stop codons before the first TOG domain, likely making them null (Fig 2B, S1 Fig). All are recessive, and homozygous mutant hermaphrodites exhibit fully penetrant embryonic lethality (Fig 2C). To investigate CLS-2 requirements, we have used the *cls-2(or1948)* allele, and hereafter we refer to oocytes from homozygous *cls-2(or1948)* hermaphrodites as *cls-2* mutants.

We first used *ex utero* live cell imaging with transgenic strains that express GFP fused to a β-tubulin (GFP::TBB-2) and mCherry fused to a histone H2B (mCherry::H2B) to examine microtubule and chromosome dynamics during meiosis I in control and *cls-2* mutant oocytes (Fig 2D; see Materials and methods). Control oocytes formed barrel shaped bipolar spindles that shortened and rotated to become perpendicular to the oocyte cortex prior to anaphase and polar body extrusion (n = 19). Consistent with previous reports [26,29,40], the meiosis I spindles in *cls-2* mutants were disorganized and lacked any obvious bipolarity, with chromosomes moving into a small cluster before failing to segregate (n = 13). Furthermore, microtubule levels appeared to be reduced, and quantification of spindle microtubule intensity over time showed a substantial reduction in microtubule levels during meiosis I anaphase compared to control oocytes (Fig 2E). These results are consistent with the established roles of CLASP family members in promoting microtubule stability (see Introduction), and with a previous report that quantified reduced microtubule fluorescence intensity at metaphase in *cls-2(RNAi)* oocytes [40]. To assess when defects in meiosis I spindle assembly first appear in *cls-2* oocytes, we used *in utero* live cell imaging and observed the early assembly of a normal cage-like microtubule structure that surrounded the oocyte chromosomes, followed by a rapid collapse of this microtubule structure to form an abnormally small cluster associated with the oocyte chromosomes (S1 Fig) (n = 5).

To further examine spindle assembly in *cls-2* mutants, we imaged meiosis I in oocytes from transgenic strains expressing an endogenous fusion of GFP to the pole marker ASPM-1 and mCherry::H2B (Fig 2F). As described previously [45], multiple small GFP::ASPM-1 foci coalesced to form a bipolar spindle in control oocytes (n = 14). In *cls-2* mutants, the GFP::ASPM-1 foci failed to coalesce to form two spindle poles but rather moved over time to form a single tight cluster of multiple small foci, and chromosomes again moved into a small cluster and failed to segregate (n = 11). We conclude that CLS-2 plays an important role in promoting microtubule stability during meiosis I and is required early for bipolar spindle assembly and chromosome segregation.

In addition to the extensive meiotic spindle defects, we also observed that *cls-2* mutants frequently failed to extrude a polar body at the end of meiosis I, consistent with previous reports [26,29]. Control oocytes regularly extruded a polar body at the end of meiosis I, as scored using transgenic strains expressing either GFP::H2B or mCherry::H2B to determine whether oocyte chromosomes remained extruded for the duration of live imaging (93 of 94 control oocytes; Fig 2G; see Materials and methods). In contrast, meiosis I polar body extrusion failed in about 84% of the *cls-2* mutant oocytes (80 of 95, Fig 2G).

## CLS-2 and MEI-1, but not spindle bipolarity, are required for polar body extrusion

Because the relationship between spindle structure and polar body extrusion is unclear, we next took a comparative approach and also examined meiosis I polar body extrusion after using RNAi to knock down either kinesin-12/KLP-18 or katanin/MEI-1, with knockdown protocols that led to fully penetrant failures in oocyte chromosome segregation (see Materials and methods). As illustrated schematically in Fig 1B, *klp-18* mutants assemble monopolar spindles while *mei-1* spindles are apolar, and both fail to segregate chromosomes (see Introduction) [33]. We first simply assessed whether polar body extrusion was successful, again using live imaging with transgenic strains expressing either GFP::H2B or mCherry::H2B fusions to score extrusion (Fig 2G). In *klp-18* mutants, chromosomes were successfully retained in a meiosis I polar body in 17 of 20 oocytes. In contrast, after MEI-1 knockdown, chromosomes were extruded and retained in a meiosis I polar body in only 5 of 27 oocytes. The absence of meiosis I polar body extrusion in *mei-1* mutant oocytes was surprising because *mei-1* mutants were originally described as typically having abnormally large polar bodies [41,42], but how often chromosome extrusion into a polar body succeeds or fails has not been reported. Our data indicate that meiosis I polar body extrusion usually fails and suggest that the abnormally large polar bodies observed in *mei-1* mutants result from defects in meiosis II, consistent with previous work [16] (see Discussion).

Based on the frequent success of polar body extrusion in *klp-18* mutants, we conclude that spindle bipolarity and chromosome segregation are not required for polar body extrusion. Thus, the failed extrusions in *cls-2* and *mei-1* mutants are not simply due to an absence of spindle bipolarity or a failure to segregate chromosomes.

## Meiotic spindle-associated furrows are abnormal in *cls-2 and mei-1* mutant oocytes

To better understand the polar body extrusion defects in oocytes lacking CLS-2 or MEI-1, we next examined the spindle-associated membrane furrows that mediate polar body extrusion in transgenic strains expressing mCherry fused to a plasma membrane marker (mCherry::PH) and GFP::H2B (Fig 3A and 3B, S2 Fig, S3 and S4 Movies). In 9 of 16 control oocytes, we observed early membrane furrows that ingressed until they pinched together to encapsulate and extrude chromosomes into the first polar body. In 6 of 16 control oocytes we observed early membrane furrows that were not as clearly resolved in our imaging data but eventually led to polar body extrusion, and in one oocyte furrows were not obvious but a polar body was nevertheless extruded.

In contrast, we observed extensive spindle-associated membrane furrowing defects in *cls-2* mutants (Fig 3C–3E, S3 and S4 Figs, S5 and S6 Movies). In 7 of 19 oocytes we observed two furrows in cross-section that retracted before pinching together, one oocyte that formed two furrows that pinched together but failed late in polar body extrusion, and one oocyte that formed two furrows and successfully extruded a polar body (S3 Fig). One of 19 oocytes

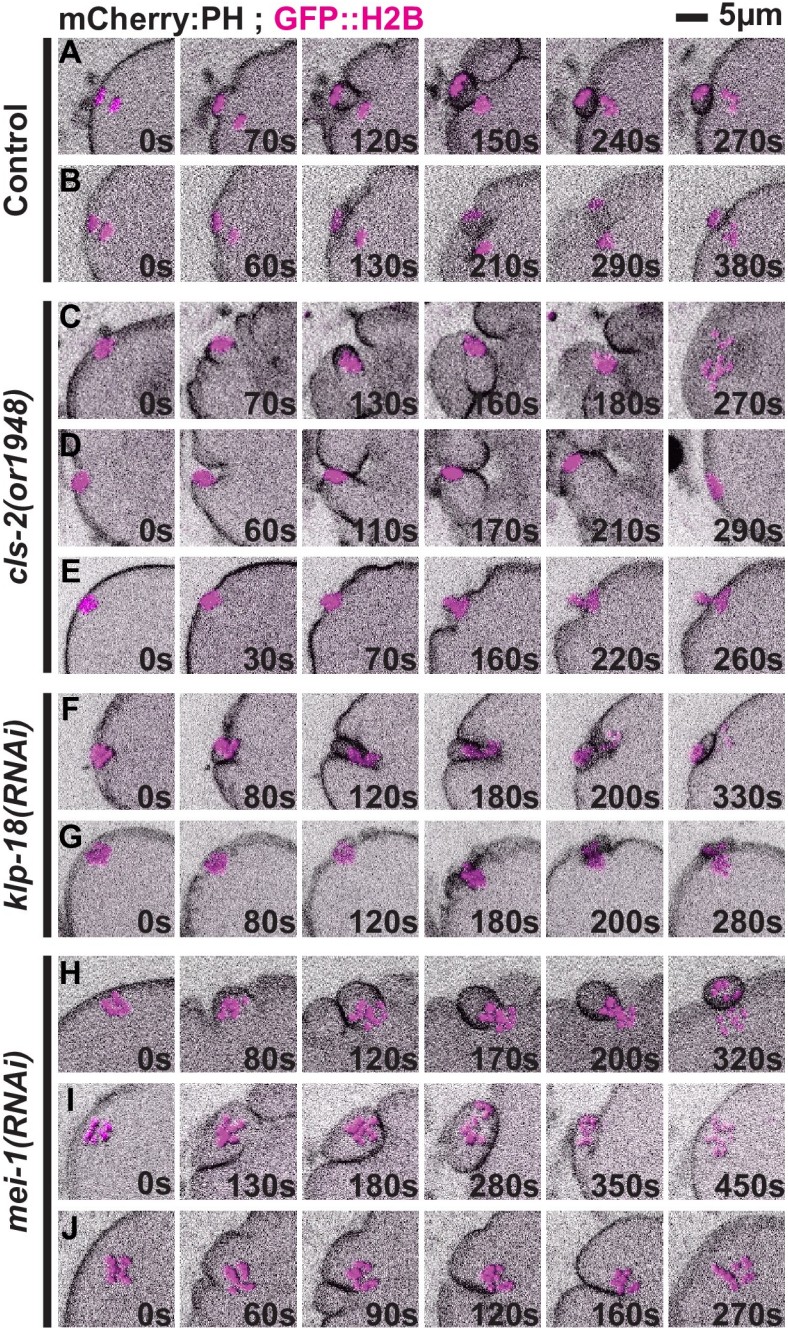

**Fig 3. Spindle-associated membrane furrowing during meiosis I in control and mutant oocytes.** Images show projections of five focal planes encompassing the chromosomes during meiosis I in oocytes expressing an mCherry fusion to a PH domain to mark the plasma membrane using sum projections and GFP::H2B using maximum projections. Representative examples of control (A, B), *cls-2* mutant (C-E), *klp-18(RNAi)* (F, G) and *mei-1(RNAi)* (H, I) oocytes. Polar body extrusion failed in C-E and H, and was successful in all others. T = 0s corresponds to the timepoint immediately before cortical furrowing begins. See text for details.

appeared to form three spindle-associated furrows in cross-section and extruded a polar body (S3 Fig). In 7 of 19 oocytes we observed only a single visible spindle-associated furrow in cross-section that ingressed either to one side of, or directly toward the oocyte chromosomes (Fig 3D, S4 Fig), suggesting that the contractile ring collapsed into a more linear ingressing

structure rather than maintaining a ring-like shape. In some cases, when the single ingressing furrow moved directly toward the oocyte chromosomes, it appeared to push chromosomes apart (S4 Fig). Such late separations of chromosomes were observed only in association with ingressing furrows that appeared to be responsible for the chromosome movement. Finally, in one oocyte the membrane dynamics were indistinct, but chromosomes were extruded into a polar body (Fig 3E), and in one oocyte there was no obvious spindle-associated furrowing and polar body extrusion failed (S4 Fig).

In oocytes depleted of *klp-18*, we observed furrows that more nearly resembled those in control oocytes (Fig 3F and 3G, S5 Fig, S7 and S8 Movies). In 4 of 10 oocytes, we observed two furrows in cross-section that ingressed and then pinched together (Fig 3F), and only 1 of these 4 failed in polar body extrusion. In 6 of 10 oocytes, we observed shallow furrows adjacent to the oocyte chromosomes (Fig 3G), and only 1 of these 6 failed to extrude a polar body.

After MEI-1 knockdown, we observed furrows that initially resembled those in control oocytes but were more widely spaced and often failed late during constriction (Fig 3H–3J, S6 Fig, S9 and S10 Movies). In 2 of 11 oocytes, we observed two furrows in cross-section that ingressed and pinched together to extrude a polar body (Fig 3H). In 3 of 11 oocytes two furrows ingressed and pinched together but then regressed and released chromosomes back into the oocyte cytoplasm (Fig 3I). In 5 of 11 oocytes two furrows ingressed but retracted before pinching together and failed in polar body extrusion (Fig 3J), and finally in 1 of 11 oocytes we observed only a single spindle-associated furrow in cross-section that failed to extrude a polar body.

To summarize, in *klp-18* mutant oocytes we observed spindle-associated furrows that usually encapsulated chromosomes and extruded polar bodies, although the oocyte chromosomes were often in close proximity to the membrane with furrows that were shallow and difficult to detect. In *cls-2* and *mei-1* mutants, meiosis I polar body extrusion frequently failed but we observed distinct defects. While membrane furrowing initially appeared relatively normal but eventually failed in most *mei-1* mutant oocytes, *cls-2* mutant furrows often appeared abnormal early during ingression and exhibited more severe defects as extrusion attempts progressed.

## Polar body contractile ring dynamics are more severely defective in the absence of CLS-2 than in *klp-18* or *mei-1* mutant oocytes

We next examined assembly and ingression of the contractile ring during oocyte meiosis I, using live cell imaging with transgenic strains expressing both a GFP fusion to the non-muscle myosin II NMY-2 and mCherry::H2B. In 11 of 11 control oocytes, NMY-2::GFP foci initially assembled into discontinuous but discernible rings over the membrane proximal pole, after spindle rotation and before extensive chromosome segregation, and then became more continuous and prominent as they ingressed and constricted between the segregating chromosomes to extrude polar bodies (Fig 4A, S7 Fig, S11 and S12 Movies).

In *cls-2* mutant oocytes, the assembly and stability of NMY-2::GFP contractile ring structures were severely defective (Fig 4B, S8 Fig, S13 and S14 Movies). In 7 of 11 oocytes, fragmented or partial contractile rings assembled and 6 of these oocytes failed to extrude a polar body, while 3 of 11 oocytes formed abnormal assemblies of NMY-2::GFP that were more linear and not ring-like, although 2 of these extruded a polar body. Finally, 1 of 11 oocytes formed a relatively normal looking contractile ring that extruded a polar body. The fragmented or partial contractile rings observed in *cls-2* mutants often collapsed into single bright foci or bands during constriction.

Contractile ring assembly and dynamics appeared much more normal in *klp-18* mutant oocytes (Fig 4C, S9 Fig, S15 and S16 Movies). In 10 of 10 oocytes after KLP-18 knockdown,

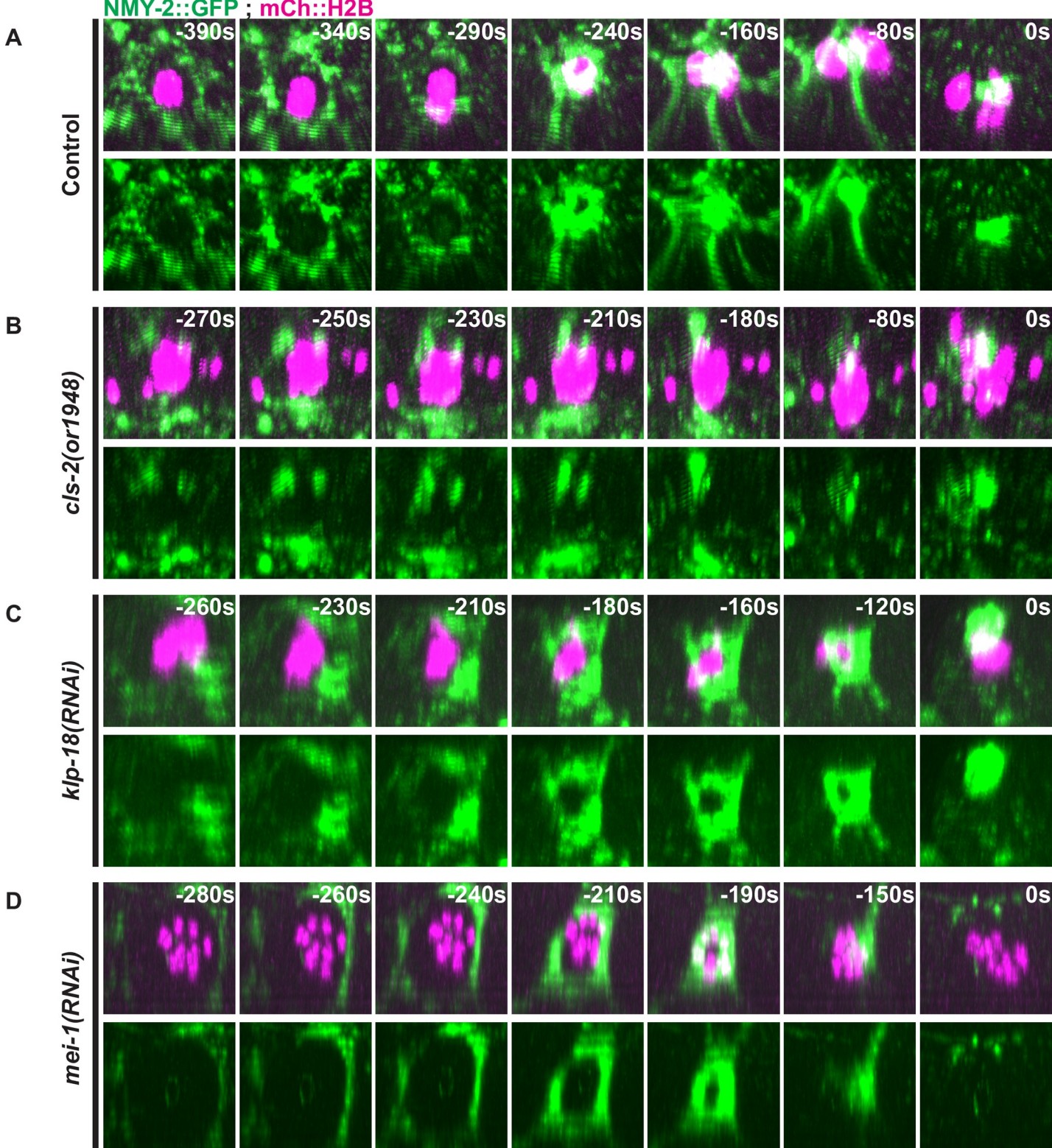

**Fig 4. Contractile ring non-muscle myosin NMY-2 dynamics during meiosis I in control and mutant oocytes.** Three-dimensionally projected and rotated images of control (A), *cls-2* mutant (B), *klp-18(RNAi)* (C), and *mei-1(RNAi)* (D) oocytes expressing NMY-2::GFP and mCherry::H2B. Z-stacks were rotated as to look down on contractile ring assembly and dynamics over time. t = 0 seconds (0s) corresponds to the end of meiosis I and beginning of meiosis II.

NMY-2::GFP foci assembled into rings that ingressed and constricted with dynamics similar to those observed in control oocytes, and in 9 of the 10 oocytes chromosomes were stably extruded into polar bodies.

In MEI-1 knockdown oocytes, ring assembly and ingression were much more normal compared to *cls-2* mutants, but we nevertheless observed a range of defects and eventual failures to extrude polar bodies (Fig 4D, S10 Fig, S17 and S18 Movies). In 11 of 13 oocytes, the NMY-2::GFP rings that initially formed appeared larger in diameter compared to control oocytes, and in 3 of these 11 oocytes the rings constricted and successfully extruded a polar body. In another 5 of these 11 oocytes, the rings constricted extensively but ultimately regressed and failed at polar body extrusion, while in 3 the rings ingressed and only constricted partially before regressing and failing to extrude polar bodies. Finally, in 2 of 13 oocytes, ring assembly and ingression were more defective and polar body extrusion failed.

To further characterize the polar body extrusion defects in *cls-2* mutants, we also examined ring assembly dynamics in transgenic strains expressing mCherry::H2B and mNeonGreen fused to the anillin ANI-1 (mNG::ANI-1), which is dispensable for assembly of the actomyosin contractile ring but required for its conversion from a ring to a tube during constriction [46] (Fig 5A, S19 and S20 Movies). In 10 of 10 control oocytes, mNG::ANI-1 assembled into contractile rings that ingressed and constricted between segregating chromosomes to extrude a polar body, while in 10 of 10 *cls-2* oocytes a fragmented contractile ring structure formed and failed to extrude a polar body. We also used two-color live imaging to examine NMY-2::mKate2 and mNG::ANI-1 simultaneously, and observed that these two contractile ring components were co-localized in both control and *cls-2* mutant oocytes (Fig 5B, S21 and S22 Movies) (n = 11 control, n = 13 *cls-2(or1948)*). We conclude that CLS-2 is required for proper ring assembly and ingression dynamics of not only NMY-2 but also ANI-1.

### CLS-2 negatively regulates membrane ingression throughout the oocyte cortex during meiosis I

Global contraction of the oocyte actomyosin cortex has been proposed to promote polar body extrusion by generating a hydrostatic cytoplasmic force that produces an out-pocketing of the actomyosin depleted membrane inside the meiotic contractile ring and pulls the membrane-tethered spindle partially into the extruded membrane pocket (see Introduction). To explore the relationship between spindle structure, global cortical contractility and polar body extrusion, we next examined membrane ingressions throughout the oocyte cortex during meiosis I in control and mutant oocytes, using transgenic strains expressing mCherry::PH and GFP::H2B fusions. To document these membrane ingressions, we used temporal overlays of a single central z-plane to portray simultaneously the membrane position at all time points throughout the period of global cortical furrowing. In control oocytes, we observed the spindle-associated furrows and a small number of additional furrows along the oocyte cortex (Fig 6A, S11 Fig) (n = 16).

In contrast, *cls-2* mutant oocytes exhibited more extensive cortical furrowing compared to control oocytes (n = 24), while oocytes depleted of either KLP-18 (n = 10) or MEI-1 (n = 14) more nearly resembled control oocytes (Fig 6A, S12 and S13 Figs, S3–S10 Movies). Quantification of global cortical furrowing showed that *cls-2* oocytes had significantly more furrows compared to control and *klp-18* oocytes (Fig 6B). We conclude that CLS-2 negatively regulates global cortical furrowing, and we suspect that the CLS-2::GFP patches detected throughout the oocyte cortex are responsible for this regulation (Fig 2A, S1 Fig, S1 and S2 Movies; see Discussion).

We next examined the dynamics of the cortical actomyosin cytoskeleton, which mediates the furrowing that occurs throughout the oocyte cortex during polar body extrusion. In control

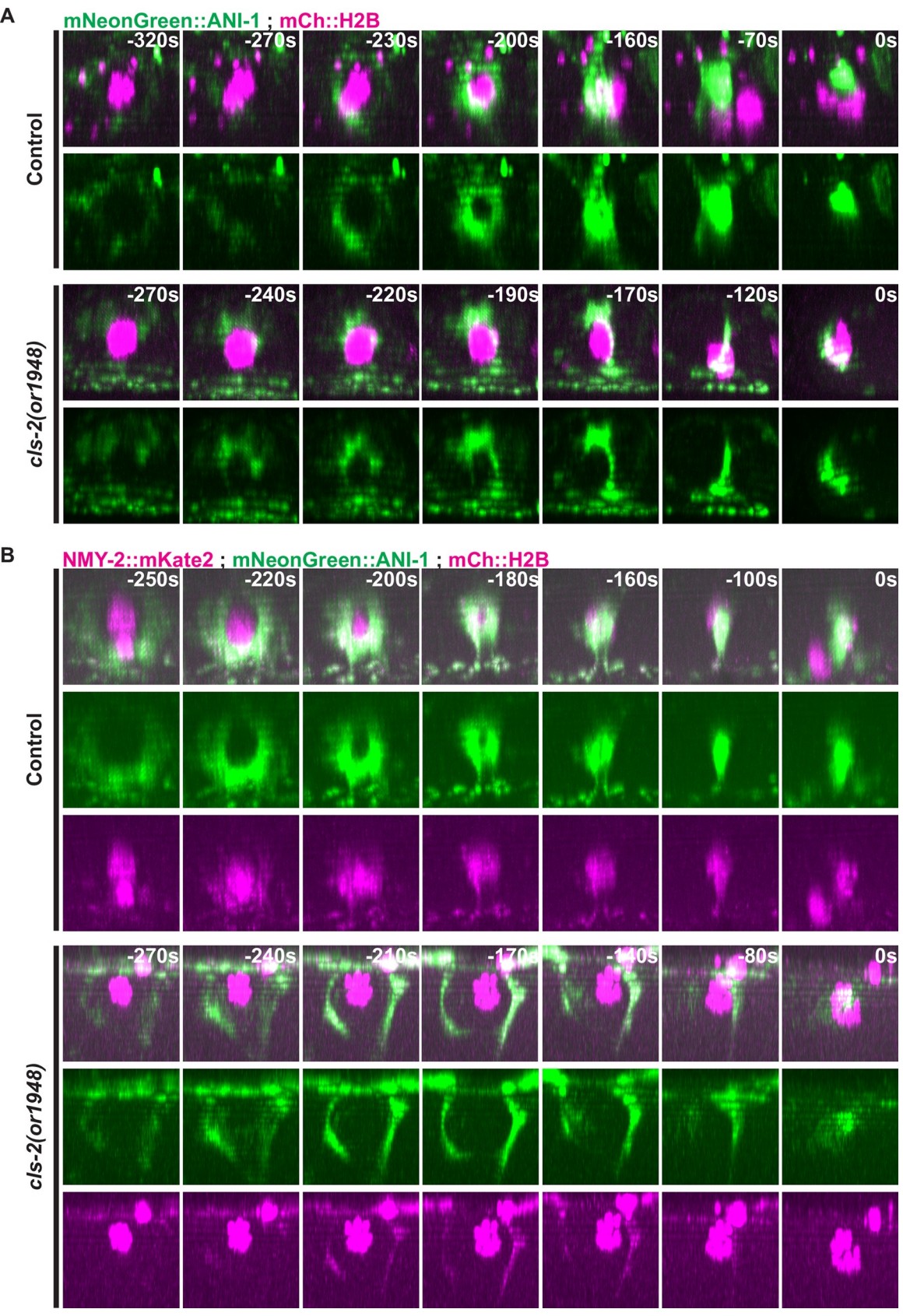

**Fig 5. Contractile ring anillin ANI-1 and non-muscle myosin NMY-2 dynamics during meiosis I in control and *cls-2* mutant oocytes.** Three-dimensionally projected and rotated images of control and *cls-2* mutant oocytes expressing mNeonGreen::ANI-1 and mCherry::H2B (A) or NMY-2::mKate2, mNeonGreen::ANI-1, and mCherry::H2B (B), with overlays shown in top row for each set. Z-stacks were rotated so as to look down on contractile ring assembly and dynamics over time. t = 0 seconds (0s) corresponds to the end of meiosis I and beginning of meiosis II.

oocytes, NMY-2::GFP (n = 11) and mNG::ANI-1 (n = 10) localized to dynamic patches throughout the oocyte cortex during meiosis I contractile ring assembly and ingression, and then dissipated late in anaphase when global cortical furrowing ends (Fig 6C and 6D, S14 and S15 Figs, S23 and S24 Movies). To determine if the increased global cortical furrowing in *cls-2* oocytes is caused by an increase in NMY-2 or ANI-1 patch size or duration, we examined the dynamics of NMY-2::GFP and mNG::ANI-1 and observed dynamics similar to those in control oocytes (Fig 6C and 6D, S16–S18 Figs, S25 and S26 Movies) and did not detect any difference in the area occupied by the cortical NMY-2::GFP or mNG::ANI-1 patches throughout the period of global cortical furrowing and polar body extrusion (S19 Fig). These data suggest that the excess global cortical furrowing observed in *cls-2* oocytes is not due to altered NMY-2 or ANI-1 patch dynamics. In contrast, the increased global cortical furrowing observed after knocking down the casein kinase CSNK-1 was associated with altered NMY-2 and ANI-1 cortical patch dynamics and with extrusion of the entire meiosis I spindle into polar bodies [15] (see Discussion).

## Loss of CLS-2 results in altered Aurora B/AIR-2 and MgcRacGAP/CYK-4 dynamics

Because (i) *cls-2* mutant oocytes fail to assemble a central spindle or segregate chromosomes, (ii) CLS-2::GFP localizes to the central spindle, and (iii) the central spindle-associated proteins Aurora B/AIR-2 and the centralspindlin component MgcRacGAP/CYK-4 are required for polar body extrusion [6,14,26,47], we next considered whether the observed polar body extrusion defects in *cls-2* oocytes might be due to altered localization of central spindle proteins to the disorganized spindle microtubules. To address this possibility, we used transgenic strains expressing GFP fusions to either AIR-2 or CYK-4 and examined their localization in control and *cls-2* mutant oocytes during the final 360 seconds of meiosis I, when anaphase chromosome segregation, global cortical furrowing, and polar body extrusion occur.

As reported previously [14,26], AIR-2 in control oocytes (n = 12) initially localized to mid-bivalent ring structures before redistributing to the central spindle during anaphase (Fig 7A). In *cls-2* mutants (n = 15), despite the lack of spindle organization, GFP::AIR-2 still localized to the mid-bivalent ring structures before redistributing throughout the dis-organized spindle as meiosis I progressed (Fig 7A). Moreover, quantification of the GFP::AIR-2 spindle to cytoplasm fluorescence ratio indicated that *cls-2* mutants have increased levels of spindle-associated AIR-2 (Fig 7B).

The centralspindlin complex member CYK-4 localized to the central spindle during anaphase in control oocytes (Fig 7C; n = 12), as reported previously [6,48]. In *cls-2* mutant oocytes, GFP::CYK-4 initially localized to the disorganized meiotic spindle, but the levels decreased rapidly over time compared to control oocytes (Fig 7C and 7D).

In summary, the central spindle proteins AIR-2 and CYK-4 show distinct alterations in their localization dynamics in *cls-2* mutant oocytes during anaphase of meiosis I. While both are still detected in association with the disorganized mutant spindle, AIR-2 levels were increased and CYK-4 levels decreased relative to control oocytes, raising the possibility that these abnormal dynamics might be at least in part responsible for the contractile ring assembly and polar body extrusion defects in *cls-2* mutant oocytes.

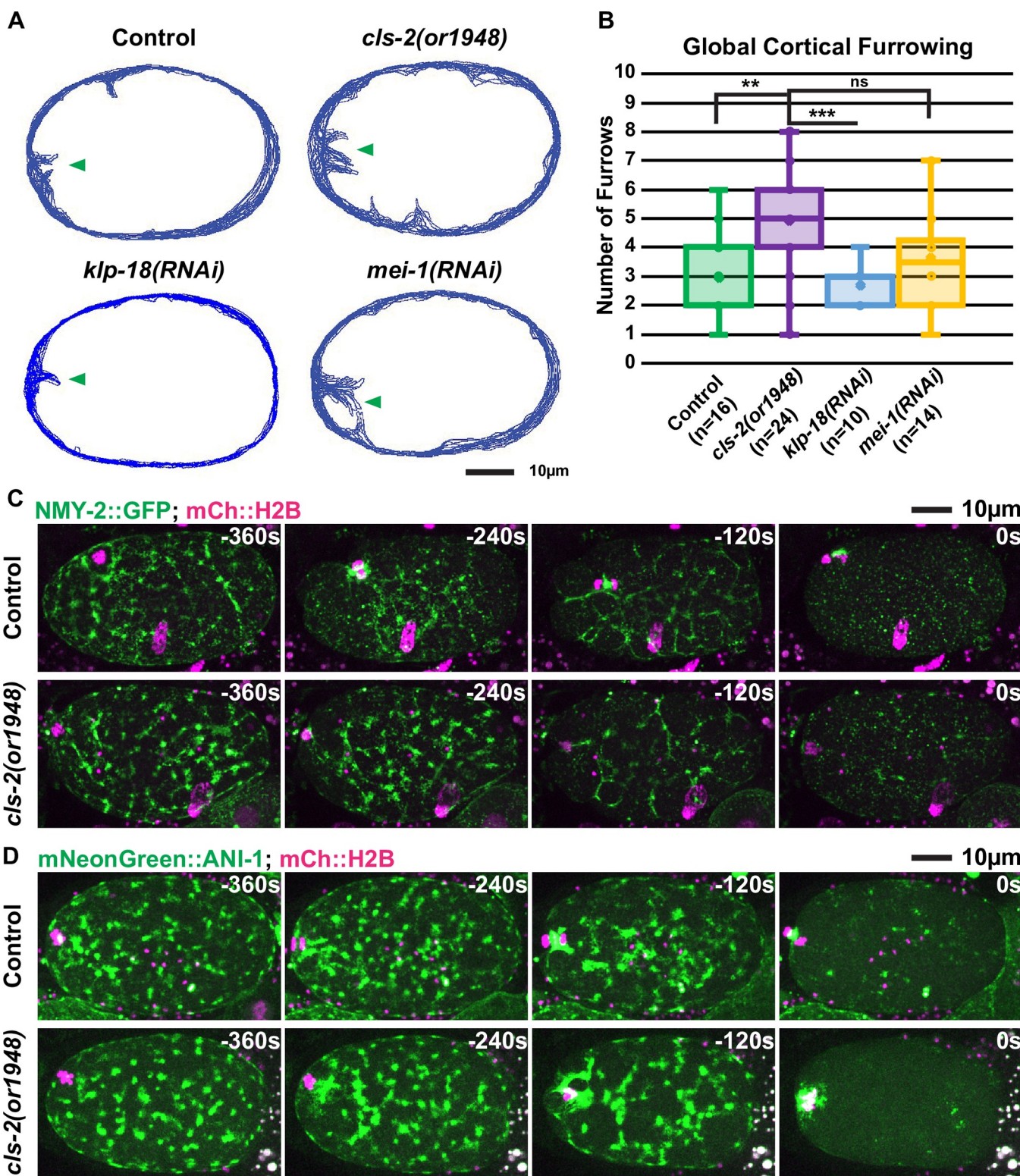

**Fig 6. Membrane ingressions throughout the oocyte cortex during meiosis I in control and mutant oocytes.** (A) Membrane temporal overlays for control, *cls-2* mutant, *klp-18(RNAi)*, and *mei-1(RNAi)* oocytes representing the membrane positions of a single focal plane during the period of meiosis I global cortical furrowing for a single oocyte of each genotype. Arrowheads indicate approximate location of the meiotic spindle and spindle-associated membrane. (B) Quantification of the number of global cortical furrows in control and mutant oocytes (see Materials and methods). t-Test results: Control vs *cls-2(or1948)* p = 0.0014 (**), *cls-2(or1948)* vs *klp-18(RNAi)* p = 3.14E-5 (***), *cls-2(or1948)* vs *mei-1(RNAi)* p = 0.054 (ns). (C and D) Control and *cls-2* mutant oocytes

expressing (C) NMY-2::GFP and mCherry::H2B or (D) mNeonGreen::ANI-1 and mCherry::H2B. t = 0 seconds (0s) corresponds to the end of meiosis I and beginning of meiosis II.

## Discussion

Remarkably little is known about the relationship between spindle structure and contractile ring assembly and constriction during oocyte meiotic cell division. To gain insight into the cues that influence contractile ring dynamics during polar body extrusion, we have examined ring assembly and constriction in three different *C. elegans* spindle assembly-defective mutants. Our results indicate that the *C. elegans* CLASP family member CLS-2 is required not only for assembly of a bipolar meiosis I spindle and chromosome segregation, but also for

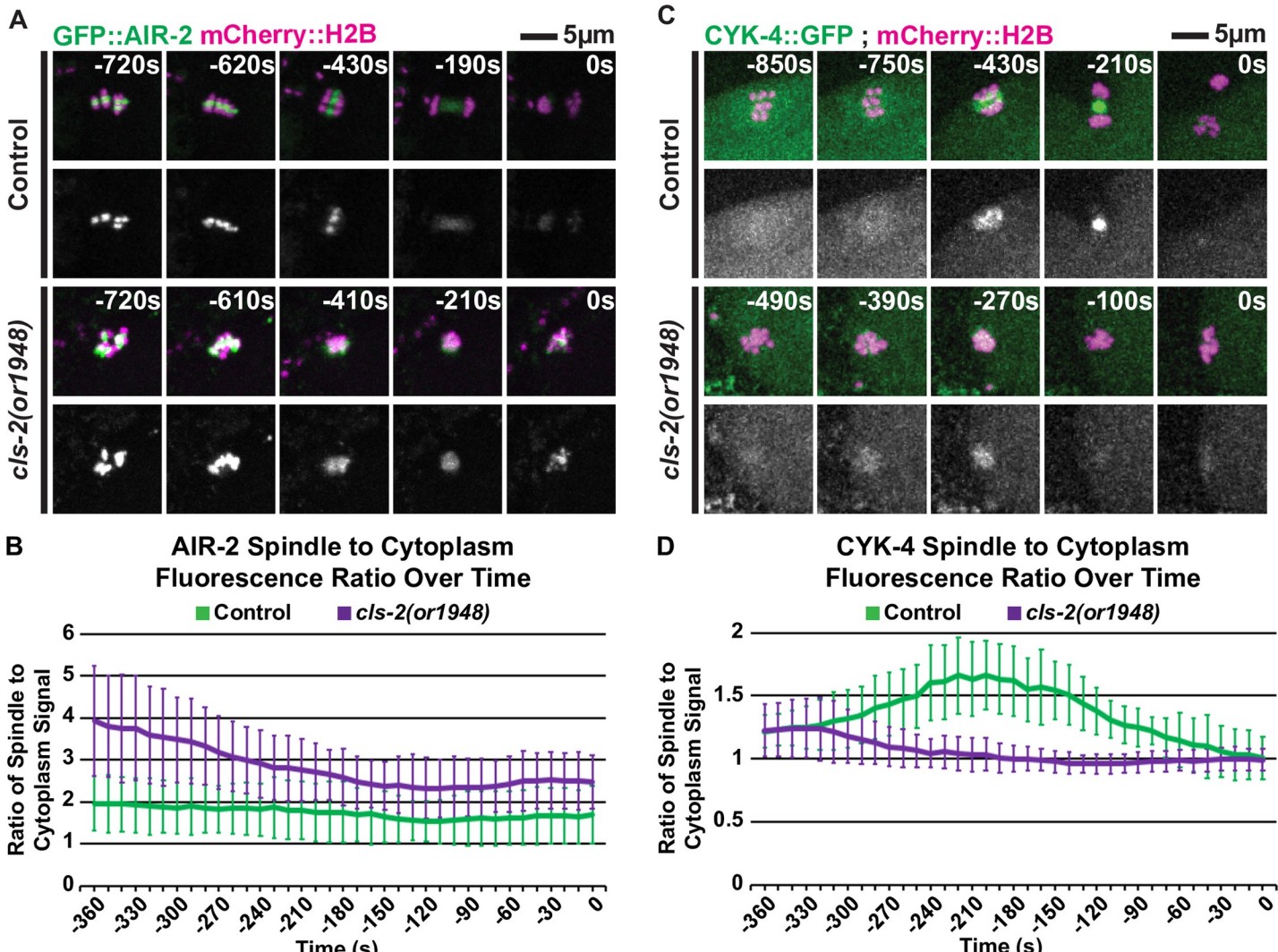

**Fig 7. *cls-2* mutant oocytes have altered AIR-2 and CYK-4 dynamics during meiosis I.** (A) Control and *cls-2* mutant oocytes expressing GFP::AIR-2 and mCherry::H2B. (B) GFP::AIR-2 spindle to cytoplasm fluorescence ratio over time for control (n = 12) and *cls-2(or1948)* (n = 12 @ -360s to -340s, n = 13 @ -330s to -310s, n = 14 @ -300s to -250s, and n = 15 @ -240s to 0s). (C) Control and *cls-2* mutant oocytes expressing CYK-4::GFP and mCherry::H2B. (D) CYK-4::GFP spindle to cytoplasm fluorescence ratio over time for control (n = 12) and *cls-2(or1948)* (n = 9 @ -360s to -340s, n = 10 @ -330s to -200s, and n = 11 @ -190s to 0s). t = 0 seconds (0s) corresponds to the end of meiosis I and beginning of meiosis II.

oocyte meiosis I contractile ring assembly or stability. This requirement for CLS-2 is not due simply to a failure in assembling bipolar spindles or to a lack of chromosome segregation, because monopolar *kinesin-12/klp-18* mutant spindles failed to segregate chromosomes but did assemble stable contractile rings that usually extruded chromosomes into a polar body. We have further shown that *mei-1/katanin* mutant oocytes, which assemble apolar and disorganized oocyte meiosis I spindles, assembled contractile rings that were abnormally large in diameter and usually failed to extrude polar bodies after some ingression, suggesting a later role for MEI-1. In addition, we observed increased cortical furrowing throughout the oocyte cortex during meiosis I in *cls-2* mutant oocytes, but not in *klp-18* or *mei-1* mutants. In mutant oocytes lacking the casein kinase CSNK-1 [15], increased global cortical furrowing is associated with altered cortical actomyosin dynamics and often results in extrusion of the entire meiosis I spindle in *C. elegans*. However, the increased cortical contractility in *cls-2* mutants was not associated with obvious alterations in non-muscle myosin dynamics and was usually associated with complete failures in polar body extrusion. We suggest that, in *cls-2* mutant oocytes, altered cortical cytoskeleton dynamics decreases cortical stiffness and thereby disrupts contractile ring assembly and furrow ingression during polar body extrusion. An alternative and not mutually exclusive possibility is that altered relative levels of central spindle proteins disrupts spindle cues that promote ring assembly and ingression.

## Using spindle assembly defective mutants to explore contractile ring dynamics during oocyte polar body extrusion

While the requirements for several factors that control oocyte meiotic spindle assembly in *C. elegans* have been described [1–3], the impact of the resulting spindle assembly defects on polar body extrusion has remained largely unexplored. Indeed, anecdotal observations have suggested that polar body extrusion can occur even in mutants with severe oocyte spindle assembly defects. For example, reducing the function of the microtubule severing complex katanin, comprised of MEI-1 and -2 in *C. elegans*, results in severely defective apolar spindles that have greatly reduced levels of microtubules and fail to organize or segregate chromosomes and yet often produce abnormally large polar bodies. Similarly, mutant oocytes lacking the kinesin-12 family member KLP-18 assemble monopolar spindles that fail to segregate chromosomes but often produce zygotes that completely lack an egg pronucleus, indicating that all of the oocyte chromosomes are sometimes extruded into polar bodies [33]. In mouse oocytes, DNA-coated beads have been shown to promote the assembly of a polar body-like cortical protrusion, which requires the small GTPase Ran that mediates chromatin signaling and, if allowed to assemble a bead-associated spindle structure, can lead to successful extrusion [17,18]. However, knockdown of the *C. elegans* Ran family member RAN-1 does not prevent chromosome segregation or polar body extrusion [19,20], suggesting that chromatin cues are not required for contractile ring assembly. Moreover, meiotic spindles that fail to translocate to the oocyte cortex induce the formation of membrane furrows that ingress deeply toward the spindle [6], and *C. elegans* mutant oocytes lacking central spindle proteins have been shown to produce abnormally large contractile rings that often fail to extrude chromosomes into a polar body [6], suggesting that the oocyte meiotic spindle does influence ring assembly and function.

To explore the relationship between oocyte meiotic spindle structure and membrane ingression during polar body extrusion, we compared polar body extrusion in three *C. elegans* mutants that are severely defective in oocyte spindle assembly. Because *cls-2* mutant oocytes failed to assemble bipolar spindles or segregate chromosomes, we also examined polar body extrusion in *klp-18/kinesin-12* mutant oocytes that assemble monopolar spindles and also fail

to segregate chromosomes [33–35]. In contrast to *cls-2* mutants, in which contractile ring assembly or stability and membrane ingression were severely defective, contractile ring assembly and ingression appeared much more normal in *klp-18* mutant oocytes, and chromosomes were usually extruded into a polar body. These findings are consistent with work in cultured mammalian cells showing that cells with monopolar mitotic spindles can still undergo cytokinesis [49]. We conclude that the defects in *cls-2* mutant oocytes are not due a failure to assemble a bipolar spindle or to segregate chromosomes, but rather reflect a more specific requirement for CLASP/CLS-2.

A specific requirement for CLS-2 is further supported by our analysis of oocyte meiotic contractile ring assembly and dynamics in *katanin/mei-1* oocytes, which assemble spindles that lack any polarity and completely fail to organize the dispersed oocyte chromosomes throughout meiosis I [33,38,39]. Furthermore, similar to *cls-2* oocytes, microtubule levels are substantially reduced in *mei-1* mutant oocytes [50]. Nevertheless, stable contractile rings usually formed in *mei-1* mutant oocytes, although the rings were larger in diameter compared to control oocytes, and we frequently observed extensive furrow ingressions that often enclosed the oocyte chromosomes but usually failed to complete constriction and regressed late in cytokinesis. In the absence of MEI-1, contractile rings can assemble and remain stable until late in meiosis I polar body extrusion, even when oocyte meiotic spindle assembly is at least as severely defective as in *cls-2* mutant oocytes.

Contractile ring dynamics during meiosis I were more normal after *mei-1* RNAi knockdown compared to *cls-2* mutants, but the furrows nevertheless often regressed and polar body extrusion usually failed. Previous studies have shown that partial loss of function mutations in *mei-1* result in abnormally large polar bodies that are produced after meiosis II, as a result of decreased microtubule severing that is required for complete disassembly of the meiosis II spindle [16,43]. Our results provide the first systematic examination of polar body extrusion during meiosis I after depletion of katanin/MEI-1, and it is possible that the late failures in polar body cytokinesis that we observed also reflect a requirement for microtubule severing. Alternatively, similar defects were observed in oocytes lacking the centralspindlin components CYK-4 and ZEN-4 [6,14,26,47], and it is possible that central spindle proteins are mis-regulated after MEI-1 knockdown. Further investigation of MEI-1 and its interactions with central spindle proteins should improve our understanding of this late requirement during polar body extrusion.

## CLS-2 regulation of oocyte cortical stiffness and contractile ring dynamics

While our analysis indicates that CLS-2 is required early for the assembly or stability of the oocyte meiosis I contractile ring, we do not know if the defects reflect a direct or indirect requirement, or how CLS-2 functions at a molecular level to promote polar body extrusion. Nevertheless, our findings lead us to suggest that CLS-2 may influence polar body extrusion by positively regulating cortical stiffness throughout the oocyte cortex. Based on our observations that (i) the increased furrowing throughout the oocyte cortex is not associated with obvious change in cortical actomyosin dynamics, (ii) CLS-2 localizes to small patches distributed throughout the oocyte cortex, and (iii) CLASP orthologs in other organisms can promote cortical microtubule attachments, we suggest that the increased cortical furrowing in *cls-2* mutant oocytes reflects a decrease in cortical stiffness, rather than an increase in cortical actomyosin contractility.

Consistent with our hypothesis that CLS-2 regulates cortical stiffness through regulation of the microtubule and possibly microfilament cytoskeleton, mutations in the *Drosophila* CLS-2 ortholog Orbit/Mast cause spermatocyte cell division defects associated with a loss of central

spindle microtubules that normally promote contractile ring assembly [51,52]. Additionally, human CLASP proteins have been shown to associate with actin filaments and may cross-link microtubules and filamentous actin [25], and mammalian CLASPs have been proposed to link microtubule plus ends with the cell cortex [53,54]. Moreover, oocyte polar body extrusion has been described as a bleb-like process, with local weakening of the actomyosin cytoskeleton inside the contractile ring promoting an out-pocketing of the membrane to form a polar body [5,6]. Studies of bleb formation in other cellular contexts have shown that microtubule destabilization can result in bleb formation [55–57]. Finally, *C. elegans* CLS-2 has been previously shown to be required for a number of microtubule dependent cortical processes in oocytes, including cytoplasmic streaming and yolk granule packing [58,59]. While it is possible that CLS-2 may act through microtubules to influence cortical stiffness, we have not yet detected significant differences in the levels or organization of cortical microtubules in *cls-2* oocytes. Higher spatial and temporal resolution imaging studies might detect subtle changes and prove informative.

Consistent with a role in regulating global oocyte cortical stiffness, we detected CLS-2::GFP in small patches, also referred to as linear elements [44] or rod-shaped structures [26], throughout the cortex in control oocytes. These patches were present early in meiosis I but dissipated before anaphase chromosome segregation, prior to initiation of the membrane ingressions that occur during anaphase in wild-type oocytes. The excessive global cortical furrowing in *cls-2* mutants may result from the loss of these cortical patches. Because the patches are undetectable at the time of furrow ingression, we suggest that CLS-2 either acts prior to the initiation of global cortical furrowing or remains present at low and undetected levels to regulate membrane ingression as furrows ingress.

CLS-2 might promote oocyte contractile ring assembly and ingression more directly, through its regulation of microtubule stability during spindle assembly. Astral microtubules associated with this acentrosomal spindle are small and limited in number, but a recent study has proposed that they nevertheless can interact, through dynein, with cortical microtubules to engage pulling forces that move the spindle toward the cortex and rotate it from its initial parallel orientation relative to the oocyte cortex to a perpendicular orientation during polar body extrusion [60]. Astral microtubule interaction with the cortex might then also influence contractile ring assembly. Consistent with such a scenario, we observed CLS-2::GFP throughout the oocyte spindle at the time when contractile ring assembly begins. Higher resolution imaging of the spindle microtubules and their proximity to the cortex in *cls-2* mutant oocytes might provide more insight.

## Complex regulation of global cortical furrowing during polar body extrusion

Comparing the consequences of reducing the functions of the casein kinase CSNK-1 and of the CLASP family member CLS-2 indicates that the relationship between global cortical dynamics and polar body extrusion is complex. CSNK-1 also limits membrane ingressions throughout the oocyte cortex during meiosis I but appears to do so through negative regulation of actomyosin dynamics, and CSNK-1 knockdown often results in extrusion of the entire meiotic spindle and all of the oocyte chromosomes into the first polar body [15]. These observations support a model in which global cortical contraction generates a hydrostatic cytoplasmic force that promotes an out-pocketing of the plasma membrane that pulls the membrane-tethered spindle pole through the contractile ring and into the forming polar body [6]. While such a mechanism may operate, it also is clear from *in utero* imaging that the contractile ring and associated plasma membrane ingress substantially prior to constricting

roughly midway along the axis of the spindle during polar body extrusion. These dynamics suggest that the spindle and the contractile ring interact to promote furrow ingression and constriction.

The different phenotypes of *csnk-1* and *cls-2* mutants indicate that negative regulation of global cortical membrane ingression during oocyte meiotic cell division may both promote and prevent the extrusion of chromosomes into polar bodies. These different outcomes presumably reflect differences in how CSNK-1 and CLS-2 influence the cortical cytoskeleton and its dynamics. CSNK-1 appears to regulate cortical actomyosin contractility, while CLS-2 might act through microtubules or microfilaments to promote cortical stiffness, with both influences being important for effective polar body extrusion.

Further investigation of cortical cytoskeleton dynamics, and the interactions of factors that regulate cortical structure and dynamics, should improve our understanding of the relationship between global cortical furrowing and oocyte meiotic contractile ring assembly. Multiple factors, including the kinesin-13 family member KLP-7, the TOG domain protein and XMAP215 ortholog ZYG-9, and Aurora A/AIR-1 have been shown to modulate cortical microtubule levels during oocyte meiotic cell division in *C. elegans* [20,48,61]. Investigation of how these different cytoskeletal regulators interact may further inform our understanding of polar body extrusion.

## Central spindle protein dynamics and polar body extrusion

The abnormal dynamics of the central spindle proteins AIR-2 and CYK-4 in *cls-2* mutant oocytes might also contribute to polar body extrusion defects. While extrusion has not been closely examined in oocytes depleted for AIR-2, the extrusion defects in oocytes lacking either CLS-2 or CYK-4 are very distinct. In *cyk-4* mutant oocytes, membrane furrows and contractile rings are abnormally large in diameter but otherwise appear normal and ingress before failing to constrict relatively late in extrusion [6]. In *cls-2* oocytes, we observed severely abnormal and unstable furrows throughout ingression, with highly defective contractile ring assembly or stability. Thus the *cls-2* defects are probably not caused by loss of CYK-4 function, even though CYK-4 levels were substantially reduced during most of anaphase in *cls-2* mutants. It seems more likely that either the increased AIR-2 levels, the altered CYK-4 dynamics at the meiotic spindle, or the altered relative levels of AIR-2 and CYK-4, might somehow disrupt contractile ring assembly. Further studies that manipulate AIR-2 or CYK-4 levels, and examine their dynamics in other spindle assembly-defective mutants, may improve our understanding of the relationship between central spindle proteins and oocyte contractile ring assembly.

## Materials and methods

### *C. elegans* strain maintenance

All *C. elegans* strains used in this study (S1 Table), were maintained at 20˚C as described previously [62].

### *cls-2* CRISPR/Cas9 allele generation

Mutations in *cls-2* were generated by injecting young adult N2 hermaphrodites with the following mixture [63]: 25μM *cls-2* crRNA (ATCAGCCGATCGACTCCGGG), 5μM *dpy-10* crRNA (GCTACCATAGGCACCAC GAG), 30μM trRNA, 2.1μg/μl Cas9-NLS, and 2.5μM *dpy-10* single-strand DNA (ssDNA, CACTTGAACTTCAATACGGCAAGATGAGAAT-GACTGGAAACCGTACCGCATGCGGTGCCTATGGTAGCGGAGCTTCACATGGCTT-CAGACCAACAGCCTAT). No homologous repair template was used for *cls-2*, and *cls-2*

DNA breaks were allowed to repair randomly. Before injection, the trRNA and crRNAs were mixed and incubated at 95˚C for 5 minutes, before cooling at room temperature for 5 minutes. After cooling, Cas9-NLS (QB3-Berkeley MacroLab) was added to the annealed trRNA and crRNAs and allowed to incubate for another 5 minutes at room temperature before the *dpy-10* ssDNA repair template was added. After injection, hermaphrodites were singled out and their broods were screened for *dpy-10* roller or dumpy co-conversion worms, which were allowed to produce broods. Those broods were then evaluated for potential *cls-2* phenotypes (embryonic lethality), and lines identified as potentially carrying mutations to *cls-2* were balanced. PCR amplified fragments were Sanger sequenced to identify the CRISPR/Cas9-induced mutations.

## Feeding RNAi Knockdown of *mei-1* and *klp-18*

RNAi knockdown of *mei-1* and *klp-18* was achieved by plating hypochlorite synchronized L1 larvae onto *E. coli* (HT115) lawns induced to express dsRNA corresponding to *mei-1* or *klp-18* [64]. Plated worms were either maintained at 20˚C until adults were imaged (*mei-1*), or maintained at 20˚C and upshifted to 26˚C 16 hours prior to imaging (*klp-18*) to ensure robust knockdown (as determined by the fully penetrant absence of chromosome segregation, indicative of the formation of monopolar meiotic spindles, during both meiosis I and II, after KLP-18 knockdown). The *mei-1* RNAi vector was from the Ahringer RNAi library [65]. The *klp-18* RNAi vector was made by amplifying a portion of the *klp-18* coding sequence from isolated N2 genomic DNA (using primers 5'-ACCGGCAGATCTGATATCATCGATGAATTCTCCA ACTTTCAA ATGCCACA-3' and 5'-ACGGTATCGATAAGCTTGATATCGAATTCCTTCG ATATGGAA GAA AGCGG-3'), which was inserted into the L4440 vector backbone using the NEBuilder HiFi DNA assembly cloning kit (NEB).

## Live-cell imaging

All imaging was carried out using a Leica DMi8 microscope outfitted with a spinning disk confocal unit–CSU-W1 (Yokogawa) with Borealis (Andor), dual iXon Ultra 897 (Andor) cameras, and a 100x HCX PL APO 1.4–0.70NA oil objective lens (Leica). Metamorph (Molecular Devices) imaging software was used for controlling image acquisition. The 488nm and 561nm channels were imaged simultaneously every 10 seconds with 1μm Z-spacing (either 16μm or 21μm total Z-stacks depending on the fluorescent markers used, with the same stack size used for all movies utilizing the same fluorescent markers).

*In utero* live imaging of oocytes was accomplished by mounting adult worms with a single row or less of embryos in 1.5μl of M9 mixed with 1.5μl of 0.1μm polystyrene Microspheres (Polysciences Inc.) on a 6% agarose pad with a coverslip gently laid over top. *Ex utero* imaging of oocytes was carried out by cutting open adult worms with a single row or less of embryos in 4μl of egg buffer (118mM NaCl, 48mM KCl, 2mM CaCl$_2$, 2mM MgCl$_2$, and 0.025 mM of HEPES, filter sterilized before HEPES addition) on a coverslip before mounting onto a 2% agarose pad on a microscope slide.

## Image analysis, quantification, and statistical analysis

General image analysis and quantification of microtubules and global cortical furrowing was carried out using FIJI software [66]. Three-dimensional projection and rotation of movies used to look at polar body contractile rings was carried out using Imaris software (Bitplane). Meiosis I polar body extrusion success was evaluated based on whether oocytes extruded any chromosomes marked by GFP or mCherry histone 2B (H2B) into a polar body that remained extruded for the period of imaging, either until meiosis I had obviously ended and meiosis II

spindle assembly began, or until pronuclei began to decondense in the one-cell stage embryo after meiosis II. We did not assess polar body extrusion during meiosis II. The end of meiosis I and beginning of meiosis II was considered to be the time at which the chromosomes left in the oocyte cytoplasm began to visibly separate from each other. Projections for spindle-associated furrow examination were made by manually isolating the 5 most spindle-associated z-planes for each time point during the period of global cortical furrowing and then sum projecting the mCherry::PH membrane signal. Membrane temporal overlays were created by overlaying the outlined membrane regions of interest for the period of furrowing (detailed below) to create a single image.

Total spindle microtubule pixel intensity was determined using the following formula: (Mean Grey Value (spindle)/Mean Grey Value (cytoplasm)) × spindle area = total spindle microtubule pixel intensity. The mean grey values for both the meiotic spindle and cytoplasm were determined by drawing a region of interest around either the meiotic spindle or a portion of oocyte cytoplasm devoid of adjacent sperm in maximum projected Z-stacks and measuring the mean grey value of the selected region in ImageJ. Spindle area was determined by measuring the area of the region of interest encompassing the meiotic spindle, or the oocyte chromosomes if the spindle could not be clearly identified (*cls-2* mutants).

AIR-2 and CYK-4 fluorescence was quantified similar to above, with the spindle to cytoplasm fluorescence ratio calculated for each time point with the following calculation: (Mean Grey Value (spindle/chromosome associated fluorescence)/Mean Grey Value (bulk cytoplasm)).

The area of cortex covered by NMY-2::GFP or mNG::ANI-1 patches over time was quantified for a subset of movies with low amounts of outside signal from nearby gonad or other embryos, as that signal interferes with image thresholding. Movies were maximum projected in FIJI, and regions of interest were drawn around each timepoint for each oocyte for the 360 seconds prior to the end of meiosis I and beginning of meiosis II (see above). A threshold was applied to each movie using either the Otsu (NMY-2::GFP) or Li (mNG::ANI-1) threshold method with dark background and stack histogram settings. Area of the NMY-2 or ANI-1 thresholded patches was then measured for the drawn regions of interest by limiting the measurements to the threshold pixels and calculating the area fraction.

Quantification of global cortical furrowing was accomplished by drawing regions of interest over the oocyte membrane signal (mCherry::PH) for a single central z-slice for the entire period of global cortical furrowing. Regions of interest were then converted to a high contrast stack of membrane positions over time, which were then analyzed using the ADAPT plugin [67] for ImageJ in order to determine curvature values across the oocyte membrane. A furrow was defined as being at least two consecutive membrane points with negative mean curvature values and a standard deviation of mean curvature at least two standard deviations above the average standard deviation of mean curvature value for the entire oocyte membrane. Membrane points fitting the criteria of a furrow (above) that were separated by a single membrane point not fitting the criteria were considered as part of the same furrow for the purposes of counting. For statistical analysis of global cortical furrowing (Fig 6B), one-way ANOVA was used to determine if there was any difference in the mean furrowing between genotypes, F-Tests to compare the variances, and two-tailed Student's t-Tests between genotypes to compare the means directly (assuming either equal or unequal variances depending on the F-Test results). All statistical analysis and graphs were completed using Excel (Microsoft).

## Supporting information

**S1 Fig. CLS-2 localizes to meiotic spindles and is required for their assembly.** (A) *In utero* time-lapse spinning disk confocal images of CLS-2::GFP and mCherry::H2B. (B) Protein

domain maps of wild type CLS-2 and CRISPR-generated *cls-2* alleles *or1949*, *or1950*, and *or1951*. Each mutation results in multiple early stop codons before the first TOG domain, with the first stop codon indicated. (C) *In utero* time-lapse spinning disk confocal images of control and *cls-2* mutant oocytes with GFP::TBB-2 and mCherry::H2B. t = 0 seconds corresponds to nuclear envelope breakdown.
(PDF)

**S2 Fig. Control oocyte spindle-associated membrane furrows.** Time-lapse spinning disk confocal images of control oocytes expressing mCherry::PH and GFP::H2B; t = 0 seconds here and in subsequent Fig 4 related supplements (S3–S6 Figs) corresponds to the time point immediately before global cortical furrowing begins, unless otherwise stated.
(PDF)

**S3 Fig. *cls-2(or1948)* oocyte spindle-associated membrane furrows.** Time-lapse spinning disk confocal images of *cls-2* mutant oocytes expressing mCherry::PH and GFP::H2B.
(PDF)

**S4 Fig. *cls-2(or1948)* oocyte spindle-associated membrane furrows.** Time-lapse spinning disk confocal images of *cls-2* mutant oocytes expressing mCherry::PH and GFP::H2B. Note that in a small number of *cls-2(-)* oocytes, we sometimes observe some separation of chromosomes. When examined in conjunction with the mCherry::PH membrane marker, such late separation of chromosomes was always associated with ingression of an oocyte meiotic cleavage furrow that appeared to push chromosomes apart (see rows 4 and 6).
(PDF)

**S5 Fig. *klp-18(RNAi)* oocyte spindle-associated membrane furrows.** Time-lapse spinning disk confocal images of *klp-18(RNAi)* oocytes expressing mCherry::PH and GFP::H2B.
(PDF)

**S6 Fig. *mei-1(RNAi)* oocyte spindle-associated membrane furrows.** Time-lapse spinning disk confocal images of *mei-1(RNAi)* oocytes expressing mCherry::PH and GFP::H2B.
(PDF)

**S7 Fig. Control oocyte NMY-2::GFP contractile rings.** Three-dimensionally projected and rotated spinning disk confocal time-lapse images of control oocytes expressing NMY-2::GFP and mCherry::H2B; t = 0 seconds in this and subsequent Fig 5 related supplements (S8–S10 Figs) corresponds to the end of meiosis I and beginning of meiosis II (see Materials and methods).
(PDF)

**S8 Fig. *cls-2(or1948)* oocyte NMY-2::GFP contractile rings.** Three-dimensionally projected and rotated spinning disk confocal time-lapse images of *cls-2* mutant oocytes expressing NMY-2::GFP and mCherry::H2B.
(PDF)

**S9 Fig. *klp-18(RNAi)* oocyte NMY-2::GFP contractile rings.** Three-dimensionally projected and rotated spinning disk confocal time-lapse images of *klp-18(RNAi)* oocytes expressing NMY-2::GFP and mCherry::H2B.
(PDF)

**S10 Fig. *mei-1(RNAi)* oocyte NMY-2::GFP contractile rings.** Three-dimensionally projected and rotated spinning disk confocal time-lapse images of *mei-1(RNAi)* oocytes expressing

NMY-2::GFP and mCherry::H2B.
(PDF)

**S11 Fig. Control oocyte membrane temporal overlays.** Control oocyte membrane temporal overlays depicting membrane positions over time at a single focal plane throughout meiosis I. Arrowheads indicate approximate location of the meiotic spindle and spindle-associated membrane.
(PDF)

**S12 Fig. *cls-2(or1948)* oocyte membrane temporal overlays.** *cls-2* mutant oocyte membrane temporal overlays depicting membrane positions over time at a single focal plane throughout meiosis I. Asterisks indicate oocytes in which polar body extrusion failed. Arrowheads indicate approximate location of the meiotic spindle and spindle-associated membrane.
(PDF)

**S13 Fig. *klp-18(RNAi)* and *mei-1(RNAi)* membrane temporal overlays.** *klp-18(RNAi) and mei-1(RNAi)* oocyte membrane temporal overlays depicting membrane positions over time at a single focal plane throughout meiosis I. Asterisks indicate oocytes in which polar body extrusion failed. Arrowheads indicate approximate location of the meiotic spindle and spindle-associated membrane.
(PDF)

**S14 Fig. Control oocyte NMY-2::GFP cortical dynamics.** Time-lapse spinning disk confocal images of control oocytes expressing NMY-2::GFP and mCherry:H2B; t = 0 seconds corresponds to the end of meiosis I and beginning of meiosis II in this and subsequent Fig 7 related supplements (S15–S18 Figs).
(PDF)

**S15 Fig. Control oocyte mNG::ANI-1 cortical dynamics.** Time-lapse spinning disk confocal images of control oocytes expressing mNeonGreen::ANI-1 and mCherry::H2B.
(PDF)

**S16 Fig. *cls-2(or1948)* oocyte NMY-2::GFP cortical dynamics.** Time-lapse spinning disk confocal images of *cls-2* mutant oocytes expressing NMY-2::GFP and mCherry::H2B. All oocytes shown succeeded in polar body extrusion.
(PDF)

**S17 Fig. *cls-2(or1948)* oocyte NMY-2::GFP cortical dynamics.** Time-lapse spinning disk confocal images of *cls-2* mutant oocytes expressing NMY-2::GFP and mCherry::H2B. All oocytes shown failed in polar body extrusion.
(PDF)

**S18 Fig. *cls-2(or1948)* oocyte mNG::ANI-1 cortical dynamics.** Time-lapse spinning disk confocal images of *cls-2* mutant oocytes expressing mNeonGreen::ANI-1 and mCherry::H2B; t = 0s corresponds to the end of meiosis I and beginning of meiosis II. All oocytes shown failed in polar body extrusion.
(PDF)

**S19 Fig. NMY-2 and ANI-1 cortical dynamics.** Graphs showing the average percent of the oocyte cortex covered by NMY-2::GFP or mNG::ANI-1 in control or *cls-2(or1948)* oocytes. Error bars show the standard deviation, and t = 0s corresponds to the end of meiosis I and beginning of meiosis II.
(PDF)

**S1 Movie. *Ex utero* CLS-2::GFP localization.** *Ex utero* time-lapse spinning disk confocal movie of a maximum projected oocyte expressing CLS-2::GFP (green) and mCherry::H2B (magenta). Frame rate is 10 frames per second.
(AVI)

**S2 Movie. *In utero* CLS-2::GFP localization.** *In utero* time-lapse spinning disk confocal movie of a maximum projected oocyte expressing CLS-2::GFP (green) and mCherry::H2B (magenta). Frame rate is 10 frames per second.
(AVI)

**S3 Movie. Control oocyte membrane furrowing.** *Ex utero* time-lapse spinning disk confocal movie of a control oocyte expressing mCherry::PH (black) and GFP::H2B (magenta). In this and subsequent oocyte membrane furrowing videos, the 5 focal planes that encompassed most of the meiotic chromosomes were used; membrane images were sum projected, histones images were maximum projected. In this and all subsequent Fig 4 related movies (S4–S10 Movies), the frame rate is 5 frames per second.
(AVI)

**S4 Movie. Control oocyte membrane furrowing.** *Ex utero* time-lapse spinning disk confocal movie of a control oocyte expressing mCherry::PH (black) and GFP::H2B (magenta).
(AVI)

**S5 Movie. *cls-2(or1948)* oocyte membrane furrowing.** *Ex utero* time-lapse spinning disk confocal movie of a *cls-2(or1948)* oocyte expressing mCherry::PH (black) and GFP::H2B (magenta).
(AVI)

**S6 Movie. *cls-2(or1948)* oocyte membrane furrowing.** *Ex utero* time-lapse spinning disk confocal movie of a *cls-2(or1948)* oocyte expressing mCherry::PH (black) and GFP::H2B (magenta).
(AVI)

**S7 Movie. *klp-18(RNAi)* oocyte membrane furrowing.** *Ex utero* time-lapse spinning disk confocal movie of a *klp-18(RNAi)* oocyte expressing mCherry::PH (black) and GFP::H2B (magenta).
(AVI)

**S8 Movie. *klp-18(RNAi)* oocyte membrane furrowing.** *Ex utero* time-lapse spinning disk confocal movie of a *klp-18(RNAi)* oocyte expressing mCherry::PH (black) and GFP::H2B (magenta).
(AVI)

**S9 Movie. *mei-1(RNAi)* oocyte membrane furrowing.** *Ex utero* time-lapse spinning disk confocal movie of a *mei-1(RNAi)* oocyte expressing mCherry::PH (black) and GFP::H2B (magenta).
(AVI)

**S10 Movie. *mei-1(RNAi)* oocyte membrane furrowing.** *Ex utero* time-lapse spinning disk confocal movie of a *mei-1(RNAi)* oocyte expressing mCherry::PH (black) and GFP::H2B (magenta).
(AVI)

**S11 Movie. Control oocyte NMY-2::GFP contractile ring dynamics.** *Ex utero* 3-dimensionally projected and rotated time-lapse spinning disk confocal movie of control oocyte

expressing NMY-2::GFP (green) and mCherry:H2B (magenta). In this and all subsequent Fig 5 related movies (S12–S18 Movies), the frame rate is 5 frames per second.
(AVI)

**S12 Movie. Control oocyte NMY-2::GFP contractile ring dynamics.** *Ex utero* 3-dimensionally projected and rotated time-lapse spinning disk confocal movie of control oocyte expressing NMY-2::GFP (green) and mCherry:H2B (magenta).
(AVI)

**S13 Movie. *cls-2(or1948)* oocyte NMY-2::GFP contractile ring dynamics.** *Ex utero* 3-dimensionally projected and rotated time-lapse spinning disk confocal movie of *cls-2(or1948)* oocyte expressing NMY-2::GFP (green) and mCherry:H2B (magenta).
(AVI)

**S14 Movie. *cls-2(or1948)* oocyte NMY-2::GFP contractile ring dynamics.** *Ex utero* 3-dimensionally projected and rotated time-lapse spinning disk confocal movie of *cls-2(or1948)* oocyte expressing NMY-2::GFP (green) and mCherry:H2B (magenta).
(AVI)

**S15 Movie. *klp-18(RNAi)* oocyte NMY-2::GFP contractile ring dynamics.** *Ex utero* 3-dimensionally projected and rotated time-lapse spinning disk confocal movie of *klp-18(RNAi)* oocyte expressing NMY-2::GFP (green) and mCherry:H2B (magenta).
(AVI)

**S16 Movie. *klp-18(RNAi)* oocyte NMY-2::GFP contractile ring dynamics.** *Ex utero* 3-dimensionally projected and rotated time-lapse spinning disk confocal movie of *klp-18(RNAi)* oocyte expressing NMY-2::GFP (green) and mCherry:H2B (magenta).
(AVI)

**S17 Movie. *mei-1(RNAi)* oocyte NMY-2::GFP contractile ring dynamics.** *Ex utero* 3-dimensionally projected and rotated time-lapse spinning disk confocal movie of *mei-1(RNAi)* oocyte expressing NMY-2::GFP (green) and mCherry:H2B (magenta).
(AVI)

**S18 Movie. *mei-1(RNAi)* oocyte NMY-2::GFP contractile ring dynamics.** *Ex utero* 3-dimensionally projected and rotated time-lapse spinning disk confocal movie of *mei-1(RNAi)* oocyte expressing NMY-2::GFP (green) and mCherry:H2B (magenta).
(AVI)

**S19 Movie. Control oocyte mNG::ANI-1 contractile ring dynamics.** *Ex utero* 3-dimensionally projected and rotated time-lapse spinning disk confocal movie of control oocyte expressing mNG::ANI-1 (green) and mCherry:H2B (magenta). In this and all subsequent Fig 6 related movies (S20–S22 Movies), the frame rate is 5 frames per second.
(AVI)

**S20 Movie. *cls-2(or1948)* oocyte mNG::ANI-1 contractile ring dynamics.** *Ex utero* 3-dimensionally projected and rotated time-lapse spinning disk confocal movie of *cls-2(or1948)* oocyte expressing mNG::ANI-1 (green) and mCherry:H2B (magenta).
(AVI)

**S21 Movie. Control oocyte NMY-2::mKate2 and mNG::ANI-1 contractile ring dynamics.** *Ex utero* 3-dimensionally projected and rotated time-lapse spinning disk confocal movie of control oocyte expressing NMY-2::mKate2 (magenta), mNG::ANI-1 (green), and mCherry:

H2B (magenta).
(AVI)

**S22 Movie. *cls-2(or1948)* oocyte NMY-2::mKate2 and mNG::ANI-1 contractile ring dynamics.** *Ex utero* 3-dimensionally projected and rotated time-lapse spinning disk confocal movie of *cls-2(or1948)* oocyte expressing NMY-2::mKate2 (magenta), mNG::ANI-1 (green), and mCherry:H2B (magenta).
(AVI)

**S23 Movie. Control oocyte NMY-2::GFP cortical dynamics.** Three example time-lapse spinning disk confocal movies of *ex utero* control oocytes expressing NMY-2::GFP (green) and mCherry::H2B (magenta). In this and all subsequent Fig 7 related movies (S24–S26 Movies), the frame rate is 5 frames per second.
(AVI)

**S24 Movie. *cls-2(or1948)* oocyte NMY-2::GFP cortical dynamics.** Three example time-lapse spinning disk confocal movies of *ex utero cls-2(or1948)* oocytes expressing NMY-2::GFP (green) and mCherry::H2B (magenta).
(AVI)

**S25 Movie. Control oocyte mNG::ANI-1 cortical dynamics.** Three example time-lapse spinning disk confocal movies of *ex utero* control oocytes expressing mNG::ANI-1 (green) and mCherry::H2B (magenta).
(AVI)

**S26 Movie. *cls-2(or1948)* oocyte mNG::ANI-1 cortical dynamics.** Three example time-lapse spinning disk confocal movies of *ex utero cls-2(or1948)* oocytes expressing mNG::ANI-1 (green) and mCherry::H2B (magenta).
(AVI)

**S1 Table. Table of *C. elegans* strains used in this study**
(PDF)

## Acknowledgments

We thank Julien Dumont, Amy Maddox, and the Caenorhabditis Genetics Center (funded by the NIH Office of Research Infrastructure Programs; P40 OD010440) for *C. elegans* strains, Jie Yang for the KLP-18 RNAi vector, Chris Doe and Diana Libuda for sharing laboratory equipment, and members of the Bowerman laboratory for helpful discussions.

## Author Contributions

**Conceptualization:** Aleesa J. Schlientz, Bruce Bowerman.

**Formal analysis:** Aleesa J. Schlientz.

**Funding acquisition:** Bruce Bowerman.

**Investigation:** Aleesa J. Schlientz.

**Methodology:** Aleesa J. Schlientz.

**Project administration:** Bruce Bowerman.

**Supervision:** Bruce Bowerman.

**Visualization:** Aleesa J. Schlientz.

**Writing – original draft:** Aleesa J. Schlientz.

**Writing – review & editing:** Aleesa J. Schlientz, Bruce Bowerman.

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
