## [Decision Letter · Decision Letter 0]

2 Jun 2020

Dear Dr Bowerman,

Thank you very much for submitting your Research Article entitled 'C. elegans CLASP/CLS-2 negatively regulates membrane ingression throughout the oocyte cortex and is required for polar body extrusion' to PLOS Genetics. We apologize for the delay as your manuscript was fully evaluated at the editorial level and by independent peer reviewers. The reviewers appreciated the attention to an important topic but identified some aspects of the manuscript that should be improved.

We therefore ask you to modify the manuscript according to the review recommendations before we can consider your manuscript for acceptance. Your revisions should address the specific points made by each reviewer. In particular: 1) make sure to include quantification (i.e. numbers, sizes....) for all the data presented in your manuscript; 2) there are several points raised by the reviewers that require further clarifications in the text, expansion of the discussion to clearly state the advancements achieved by the current study, and inclusion of proper citations; and 3) include labeling of supplementary figures.

[LINK]

Yours sincerely,

Mónica P. Colaiácovo

Associate Editor

PLOS Genetics

Gregory P. Copenhaver

Editor-in-Chief

PLOS Genetics

Reviewer's Responses to Questions

**Comments to the Authors:**

Reviewer #1: In the present manuscript, Schlientz & Bowerman investigate the relationship between spindle assembly and polar body extrusion in C. elegans oocytes. Using novel alleles of the CLASP orthologue CLS-2, the authors show that CLS-2 and katanin (MEI-1) play different roles during polar body extrusion. I particularly liked the idea that late meiosis I events might not be completely linked to early ones. As the authors rightly point out, this is a hard topic to tackle because of early phenotypes obscuring the interpretation of the late ones. However, with a good combination of an accurate description of the results together with the combination of in utero and ex utero experiments, the authors provide very interesting insight into the different roles of key proteins during meiosis, specially CLS-2. Importantly, the experiments are performed using live oocytes, allowing for a proper dynamic assessment of spindle assembly, segregation, and polar body extrusion.

I was very impressed by the level of detail and transparency with which the phenotypes are described and the work the authors took to include all these in the supplementary material.

Overall, I think this work, together with the other recent preprint from the lab, provides very interesting insights into the dynamics of spindle assembly and polar body extrusion in C. elegans oocytes. I would therefore favour publication in PLOS Genetics after some revisions, listed below.

1) In Figure 2E, why does the tubulin quantif. only go back to -340 secs while the images shown go back to -840 sec?

2) Figure 3 needs some quantification. Specially in the case of CYK-4, the wild type and mutant look very different and the text only makes a general remark about this.

3) Specifically, in figure 3A it appears that CLS-2 depletion leads to a phenotype in which there seems to be some degree of chromosome segregation, but then polar body fails. While I am not sure how reproducible this phenomenon is, it would make a good case for stating that polar body extrusion can’t happen even when anaphase is not completely affected.

4) In general, the use of this new cls-2 allele is a nice addition that will also prove useful for other studies. As such, it would be good to have a comparison with RNAi and/or auxin-inducible degradation, because the allele used could have the downside of chronic CLS-2 depletion (i.e. early meiosis defects?).

5) One slightly confusing issue is how ‘time = 0’ is defined in the cls-2 mutant. While I acknowledge this will never be an easy task (we routinely have the same issue), the reader would benefit from some clarification to know that the time scales in some experiments are only good for internal reference of each movie and not to be used for comparison between movies.

6) We felt that the data on AIR-2 and CYK-4 really needs some quantitation. Specially in the case of CYK-4, the wild type and mutant look very different and the text only makes a general observation from this.

7) Is the klp-18(RNAi) phenotype 100% penetrant (i.e. are all spindles monopolar)? If so, then disregard. If it isn’t, it would make sense to explicit that the 17/20 cases with PB extrusion come from monopolar spindles and it is not a consequence of partial depletion.

8) ‘Based on the frequent success of polar body extrusion in klp-18 mutants, we conclude that spindle bipolarity and chromosome segregation are not required for polar body extrusion.’ This could be a bit of a stretch, since there is a form of segregation-like chromosome movement that could, in theory, provide with whatever mechanism is necessary for PB extrusion. Here again, as mentioned in 1), If you get to see a phenotype with attempt segregation followed by failed PB extrusion, that would be very useful, as it would prove your case.

9) For the NMY-2 and ANI-1 movies (which I have to admit I am no expert), it would be helpful to have a measure of chromosome segregation (as a reference only) together with time reference.

10) How do the other steps look in the cls-2 mutants that do extrude a PB? Spindle assembly, chromosome alignment, segregation, etc…? This relates to question 3). Could the authors compare spindle assembly defects in the 15/95 cls-2 mutants movies that do form a polar body vs the 80/95 that don’t, using a Fisher test maybe?

Reviewer #2: This manuscript is meant to address the mechanism of polar body formation during C. elegans meiosis. This is a significant problem because polar body formation is conserved in most animal species and because polar body failure results in polyploidy and death. Specifically this manuscript provides high quality phenotypic analysis of polar body formation by time-lapse imaging in a new CRISPR knockout of the microtubule-binding protein CLS-2 and in RNAi depletions of two other microtubule regulators, KLP-18 and MEI-1. Although the results are of high quality, most of the results have either been published previously [polar body failure in cls-2(RNAi) complemented with various RNAi-resistant point mutations (Laband 2017)] or are negative [no change in centralspindlin or acto-myosin in the cls-2 knockout]. In addition, there is no mechanistic hypothesis stated and there is no clear mechanistic conclusion drawn from the results. The use of “we explore” in the final sentence of the abstract suggests that no specific question was asked and that no definitive conclusion was reached in the paper. These issues could potentially be addressed by extensive rewriting of the manuscript. This would require stating a specific question or hypothesis and explaining how the results answer this question and how previously published results could not answer this question.

Detailed comments below are meant to help clarify the significance of the work to someone reading the manuscript.

Line 65: “During anaphase the contractile ring ingresses past both the membrane-proximal pole…” The only reference for this statement is Fig. 1 which is a cartoon with no reference to data in the legend. This statement is also made before the text has narrowed down to C. elegans as a species. In many organisms without an eggshell, the spindle pushes out through the contractile ring. In C. elegans in hyperosmotic conditions where the plasma membrane is pulled away from the eggshell, the spindle also pushes out (Dorn et al., 2011). In utero, the plasma membrane starts pressed against the eggshell and the contractile ring ingresses inward (Fabritius et al., 2011) as stated here. This statement needs to be clarified for both accuracy and proper citation.

Line 150: The nature of the CLS-2::GFP should be stated (CRISPRd?, complementing RNAi?) either here or referenced to someplace that it is stated.

Line 154, Fig. 2A: It is stated that metaphase-specific patches of CLS-2::GFP on the entire cortex might suggest that CLS-2 acts on the entire cortex. If this suggestion is going to be stated, it should be clarified whether the patches correspond to linear elements, myosin patches or perhaps clusters of cortical microtubules. If it is not possible to figure this out because of COVID-19, then removing the suggestion that this localization is indicative of global cortical activity might be appropriate.

Line 160: There should be a citation at the end of the first sentence.

Line 166: “fully penetrant maternal-effect embryonic lethality” would be clearer.

Line 194: What is new with the microtubule and chromosome phenotype? The results with the newly generated null allele appear to confirm the previously published RNAi phenotypes which were complemented with RNAi-resistant transgenes in Laband 2017. The new results are thus a second control showing that the phenotypes are not due to off target RNAi effects. The only thing these results show are that Dumont 2010 and Laband 2017 were likely looking at complete rather than partial RNAi depletion. I actually think it is important to reproduce the RNAi work of others using a CRISPR knockout but this needs to be clearly explained to the reader.

Line 201: What is new about the polar body phenotype? It seems like less of an advance than the Laband 2017 results which included polar body phenotypes with point mutations in specific TOG domains.

The description of the assay for polar body extrusion failure is in the Methods and makes it unclear whether there is a defect in extrusion of the second polar body. If this information can be gleaned from existing data, this would at least be a slight advancement over previously reported polar body failure phenotypes. Line 240 speculates that mei-1 mutants might have a meiosis II polar body defect which suggests that the time-lapse imaging in this study did not go through completion of meiosis II. This manuscript thus does not resolve the previously published large polar body phenotype with the current no polar body phenotype for mei-1. A standard in this field would be to report polar body number and size in mitotic embryos. This would be especially important for klp-18(RNAi) which is reported to succeed in polar body formation. An-1 knockdown polar bodies do not reveal their failure until very late. This manuscript also uses RNAi by feeding for mei-1 whereas previous studies have used allelic series of mei-1.

The AIR-2 and myosin dynamics results are all negative. Some negative results would be fine if a positive result was reported that yielded some clue as to why cls-2 knockouts do not make polar bodies.

Line 244: The conclusion that spindle bipolarity and chromosome segregation are not required for polar body extrusion because 15/20 klp-18(RNAi) embryos successfully extruded a polar body (Fig. 2G) could be a significant result but is only weakly supported. The successful polar body extrusion could be due to partial RNAi. Filming with transgenes (GFP::ASPM-1) that convincingly showed a lack of spindle bipolarity during late anaphase would be required to address this issue. Mullen and Wignall (2017) presented evidence that klp-15/16-depleted embryos have no bipolarity at metaphase but assemble a midzone during anaphase. Filming GFP:SPD-1 in klp-18(RNAi) might resolve whether this is happening during klp-18(RNAi) polar body formation. I am not suggesting that the authors do experiments during quarantine. I am just pointing out the brevity of the results supporting this conclusion in the manuscript. The novelty/lack of novelty of monopolar spindle polar body formation could be cited more extensively to Canman et al. (2003) and others have shown that monopolar spindles can induce mitotic cleavage furrows but there is no discussion of this.

Regarding the conclusion that CLS-2 directly suppresses global cortical furrowing. CLS-2 depletion reduces microtubule density in the spindle (this study) and CLS-2 depletion blocks microtubule-dependent cytoplasmic streaming (Yang et al., 2003) and microtubule-dependent yolk granule packing (McNally et al., 2010) throughout the embryo. There should be a clearer discussion of the possibility that reduced density of cortical microtubules causes the global increase in furrowing. Microtubule-dependent suppression of cortical furrowing has been reported many times including tubulin(RNAi) in C. elegans meiosis (Fabritius et al., 2011) and during mitosis and could be more clearly discussed. This is an area that could be developed into a hypothesis that is stated at the beginning of the paper.

Reviewer #3: Schlientz and Bowerman addressed the relationship between meiotic spindle assembly and polar body extrusion. They characterized three spindle assembly mutants, cls-2, klp-18, and mei-1. Mutants of CLS-2 fail to form bipolar spindles, had low levels of microtubules, and failed to segregate DNA. Previous work showed that mutants of KLP-18 form monopolar spindles and fail to segregate chromosomes. Mutants of MEI-1 form apolar spindles and fail to segregate chromosomes. Mutants of CLS-2 mostly do not extrude polar bodies. Live imaging shows that there are defects in ingression and they have abnormal assembly of the NMY2:::GFP contractile rings. The mei-1 mutants mostly do not form polar bodies, but the defect is later than in cls-2 mutants. The oocytes initially ingress but fail later. Many of the contractile rings assemble and initiate constriction but then fail. The klp-18 mutants mostly do form polar bodies, suggesting that bipolar spindle formation and chromosome segregation is not required for polar body formation. In the absence of CLS-2, there were more global cortical furrows, suggesting that CLS-2 negatively regulates furrowing.

This manuscript is mainly descriptive, and at times this can be a bit frustrating. For example, there are many defects in the cls-2 mutants and it is difficult to discern whether the later defects in contractile ring assembly are due to earlier defects in ingression. Or, even in understanding how these regulators of spindle assembly affect these processes and why the phenotypes are so variable.

But, overall the live cell analysis of the steps of polar body formation provide a nice and important understanding of how/when the processes fail in the different mutants. And, this study provides a foundation for future more mechanistic studies. The major findings of this study include: i) spindle bipolarity and chromosome segregation are not required for polar body contractile ring formation and chromosome extrusion; ii) CLS-2 is required for early contractile ring assembly; iii) MEI-1 is needed for full contractile ring constriction; and, iv) CLS-2 negatively regulates membrane ingression.

Overall, I have only minor suggestions for improvements to the manuscript:

1) The authors propose that CLS-2 prevent membrane ingressions at the oocyte cortex. Do the CLS-2::GFP patches co-localize with actin filaments?

2) Can the authors provide the data (in graph form) showing the dynamics of NMY-2::GFP and mNG::ANI-1 in control and mutant oocytes?

3) It seems interesting that the mutants can MEI-1 mutants can initiate polar body formation and then regress back to no polar body formation. Can the authors further speculate on the regulation of how this occurs?

4) For clarity, it would greatly help the reader if the figures were labeled with the mutants that are being described so that the reader does not need to refer back to the figure legends. This has been done in the main figures, but not in most of the supplementary figures (Figures S3- S12, S14-18).

5) In Figures 7A and S12-S13, it would be helpful to point out where the spindle was localized.

6) Correct typo on line 533.

**Have all data underlying the figures and results presented in the manuscript been provided?**

Reviewer #1: Yes

Reviewer #2: Yes

Reviewer #3: Yes

PLOS authors have the option to publish the peer review history of their article (what does this mean?). If published, this will include your full peer review and any attached files.

Reviewer #1: Yes: Federico Pelisch

Reviewer #2: No

Reviewer #3: No

---

## [Editor Report · Decision Letter 1]

10 Aug 2020

Dear Dr Bowerman,

We are pleased to inform you that your manuscript entitled "C. elegans CLASP/CLS-2 negatively regulates membrane ingression throughout the oocyte cortex and is required for polar body extrusion" has been editorially accepted for publication in PLOS Genetics. Congratulations!

Yours sincerely,

Mónica P. Colaiácovo

Associate Editor

PLOS Genetics

Gregory P. Copenhaver

Editor-in-Chief

PLOS Genetics

Comments from the reviewers (if applicable):

**Data Deposition**

http://datadryad.org/submit?journalID=pgenetics&manu=PGENETICS-D-20-00475R1

**Press Queries**

---

## [Editor Report · Acceptance letter]

28 Sep 2020

PGENETICS-D-20-00475R1 

C. elegans CLASP/CLS-2 negatively regulates membrane ingression throughout the oocyte cortex and is required for polar body extrusion 

Dear Dr Bowerman, 

We are pleased to inform you that your manuscript entitled "C. elegans CLASP/CLS-2 negatively regulates membrane ingression throughout the oocyte cortex and is required for polar body extrusion" has been formally accepted for publication in PLOS Genetics! Your manuscript is now with our production department and you will be notified of the publication date in due course.

With kind regards,

Jason Norris

PLOS Genetics

On behalf of:
